# Nominally identical microplastic models differ greatly in their particle-cell interactions

Simon Wieland [1,2,7], Anja F. R. M. Ramsperger[1,2,7], Wolfgang Gross[1,7], Moritz Lehmann [3], Thomas Witzmann [4], Anja Caspari[4], Martin Obst[5], Stephan Gekle [3], Günter K. Auernhammer [4], Andreas Fery [4,6], Christian Laforsch [2,8] ✉ & Holger Kress [1,8] ✉

Due to the abundance of microplastics in the environment, research about its possible adverse effects is increasing exponentially. Most studies investigating the effect of microplastics on cells still rely on commercially available polystyrene microspheres. However, the choice of these model microplastic particles can affect the outcome of the studies, as even nominally identical model microplastics may interact differently with cells due to different surface properties such as the surface charge. Here, we show that nominally identical polystyrene microspheres from eight different manufacturers significantly differ in their $\zeta$-potential, which is the electrical potential of a particle in a medium at its slipping plane. The $\zeta$-potential of the polystyrene particles is additionally altered after environmental exposure. We developed a microfluidic microscopy platform to demonstrate that the $\zeta$-potential determines particle-cell adhesion strength. Furthermore, we find that due to this effect, the $\zeta$-potential also strongly determines the internalization of the microplastic particles into cells. Therefore, the $\zeta$-potential can act as a proxy of microplastic-cell interactions and may govern adverse effects reported in various organisms exposed to microplastics.

The first observation of microscopically small plastic particles in the ocean was made by Carpenter et al. in 1972[1], and 50 years later, plastic particles were detected in all environmental compartments[2]. In 2004, Thompson et al. coined the term microplastics, defined as particles smaller than 5 mm[3,4]. The abundance of microplastics in the environment is associated with potential risks for environmental and human health[5,6]. Organisms are predominantly exposed to microplastics via inhalation or ingestion. The latter has already been described for a variety of organisms ranging from protozoans[7] to even vertebrates[8,9]. Upon ingestion or inhalation, microplastic particles can translocate from the gastrointestinal tract or the respiratory organs into the circulatory system[6,10] and surrounding tissues, leading to adverse effects such as inflammatory responses[9,11]. Here, the cellular internalization of microplastic particles is a potential pathway for the translocation into tissues[12]. The cellular internalization of microplastic particles was reported for pristine particles[13,14] as well as environmentally exposed

[1]Biological Physics, University of Bayreuth, Bayreuth, Germany. [2]Animal Ecology I and BayCEER, University of Bayreuth, Bayreuth, Germany. [3]Biofluid Simulation and Modeling – Theoretical Physics VI, University of Bayreuth, Bayreuth, Germany. [4]Leibniz Institut für Polymerforschung Dresden e. V., Institute of Physical Chemistry and Polymer Physics, Dresden, Germany. [5]Experimental Biogeochemistry, BayCEER, University of Bayreuth, Bayreuth, Germany. [6]Physical Chemistry of Polymeric Materials, Technische Universität Dresden, Dresden, Germany. [7]These authors contributed equally: Simon Wieland, Anja F. R. M. Ramsperger, Wolfgang Gross. [8]These authors jointly supervised this work: Christian Laforsch, Holger Kress. ✉e-mail: christian.laforsch@uni-bayreuth.de; holger.kress@uni-bayreuth.de

particles coated with an eco-corona[15]. Among other cell types, a focus was set on macrophages, since in many organ systems, such as the lungs, macrophages are among the first cells to encounter inhaled or ingested microplastic particles[6,16]. Furthermore, due to the mobility of these cells they can act as transporters for microplastic particles that translocate them into tissues and lead to their distribution in the organism[6].

To date, the predominantly used polymer in microplastics research is polystyrene[17,18] and the vast majority of studies was conducted with monodisperse, spherical polystyrene particles[19–22]. For studies that use the same polymer type, shape, and size range, one should expect that the results are comparable and consistent with each other. However, studies on potential negative effects of microplastics on organisms show a large variety of sometimes seemingly contradictory results. For instance, negative effects, such as a reduction in metabolism and gamete production, inhalation toxicity, inflammation, and oxidative stress were found in oysters[23], rats[24], and mice[25]. In contrast, no such negative effects were found in other studies in barnacle larvae[26] and mice[13]. On the cellular level, similar discrepancies have been observed. For example, studies using spherical polystyrene particles in the micrometer size range showed that microplastic particles were readily internalized by the cells, inducing an increase in reactive oxygen species and cytotoxic effects[14,27]. In contrast, another study using similar particles observed that only a minor fraction of microplastic particles were internalized by cells, causing no or only little cytotoxicity[13].

Current microplastics research is based on particles produced by a large number of manufacturers[14,27–31]. Although these particles are all sold as polystyrene microspheres, particles from different commercial sources can significantly differ in their physicochemical properties. Ramsperger et al. showed that two types of polystyrene particles without a dedicated surface functionalization differed in their monomer content, ζ-potential, and surface charge densities, leading to differences in metabolic activity and cell proliferation[32].

Especially the ζ-potential, which is the electrical potential at the shear plane of a particle in a suspension[33], has been discussed to influence the particle-cell interactions and the internalization[6,32,34]. For nanoparticles, it is a well-established fact that cellular interactions (including internalization) and cytotoxicity depend on the particle's surface charge and the ζ-potential[35–38]. While neutral nanoparticles only minimally interact with cells[35,36], positively charged nanoparticles interact strongly with both phagocytic and non-phagocytic cells[36,37], whereas negatively charged nanoparticles interact more frequently with phagocytic cells[35]. Additionally, the mechanism of internalization seems to depend on the polarity and density of the nanoparticles' surface charge[35,36]. However, not only the physicochemical properties of the nanoparticles, but also the cell type seems to affect the internalization of nanoparticles into cells[35,36].

Although the role of surface charge and ζ-potential for cellular interactions and internalization is well-known for nanoparticles, research findings for microparticles are less unanimous. On the one hand, studies with microparticles are not conclusive about the role of their surface charge and ζ-potential for their cellular interactions and internalization. Since cells generally possess a net negative ζ-potential[39], it is expected that microparticles with a net positive ζ-potential interact with cells more frequently and become internalized more often[40]. This has indeed been observed for polylactic acid (PLA), polylactide-co-glycolic acid (PLGA), and polyethylene oxide/polylactic acid block copolymer (PELA) microparticles, where less negatively charged microparticles adhere stronger to cells and become internalized more often[41,42]. In contrast, other studies showed that both negatively and positively charged microparticles are phagocytosed efficiently[43], and an increase in negative surface charge can lead to an increase in internalization efficiency[43,44].

On the other hand, results from nanoparticles cannot simply be transferred to microparticles:[45–47] For example, due to their different surface-to-volume ratio, nanoparticles and microparticles interact differently with cells and tissues[47,48]. Furthermore, the mechanisms of cellular internalization strongly differ between nanoparticles and microparticles. Nanoparticles can be internalized by cells via a number of different endocytic pathways, including clathrin-mediated endocytosis, caveolin-mediated endocytosis, macropinocytosis, and passive transport into cells[35,47]. However, due to their size, internalization of microparticles is limited to the actin-dependent processes of phagocytosis and macropinocytosis[47,49,50].

The effect of the ζ-potential of microplastic particles on their interactions with cells and organisms is even less clear. Of 216 studies about possible effects of microplastics for aquatic or mammalian models currently listed in the ToMEx database, only 17% provided the ζ-potential of the microplastic particles[51]. Furthermore, despite the indications for a role of the ζ-potential for cellular interactions and the seemingly contradictory results in microplastic cytotoxicity studies, the role of the ζ-potential has not yet been systematically investigated for otherwise identical microplastic particles. Furthermore, it was shown that environmental exposure, leading to the formation of an eco-corona on the particles, alters particle-cell interactions[15]. However, it is not clear whether these changes in particle-cell interactions are caused by changes in the ζ-potential.

To shed light on the role of the ζ-potential as one driver for microplastic-cell interactions, we investigated nominally identical polystyrene particle types from eight different manufacturers (detailed information and subsequent abbreviations see Table 1). Next to the pristine microplastic particles we additionally incubated spherical PS-particles from MM in salt and freshwater to investigate the influence of the environmental exposure on the ζ-potential. We measured the particles' ζ-potential with a zetasizer for each particle type and developed a single-cell single-particle multiplexed microfluidic platform with an artificial intelligence-based data analysis to quantify the particle-cell adhesion strength. Furthermore, we measured the proportion of internalized microplastic particles for each particle type by confocal microscopy. In this way, we aim to quantify how the ζ-potential of nominally identical microplastic particles, which may be additionally altered by exposure to environmental media, affects their binding kinetics, adhesion strength, and cellular internalization probability.

## Results
### The ζ-potential of nominally identical microplastics differ

Although nominally identical, the microplastic particles from the different manufacturers were different in scanning electron micrographs. There were differences in their equivalent diameter, eccentricity, and surface roughness (Fig. 1, Supplementary Note 1, Supplementary Table 1). Furthermore, the ζ-potential of the particles from different manufacturers varied from −93.1 mV (ST) to −4.7 mV (MM) (Table 1). ST (−93.1 mV) and PY (−83.8 mV) had similar strongly negative ζ-potentials whereas TJ (−45.5 mV) had a medium ζ-potential. All other pristine particles had ζ-potentials closer to zero: TS, MG, PX, KI, MM (−13.7 mV, −12.6 mV, −7.5 mV, −5.3 mV, and −4.7 mV, respectively).

Incubation of the microplastic particles in cell culture media led to a decrease in the magnitude of their ζ-potential. However, the ζ-potential after incubation of the microplastic particles in cell culture media was strongly correlated to their initial ζ-potential (Pearsons's $R = 0.8$, $P = 0.004$): particle types that were strongly negative initially were still strongly negative after incubation, and particle types that had an initial ζ-potential close to zero were still almost neutral after incubation in cell culture media (Supplementary Table 1, Supplementary Fig. 1). All particle types showed a high colloidal stability in the cell experiments, no significant aggregation of particles occurred (Supplementary Fig. 2)

**Table 1 | Specifications of all polystyrene microparticles**

| Sample | Manufacturer | Product name | Product no. | Modification | Nominal diameter (µm) | Measured diameter (µm) | ζ (mV) |
|---|---|---|---|---|---|---|---|
| PY | Polysciences, Inc. | Polybead® Microspheres | 17134-15 | None | 3.00 | 3.08 ± 0.2 | −83.8 ± 0.3 |
| MM | Micromod GmbH | micromer® | 01-00-303 | None | 3 | 2.94 ± 0.02 | −4.7 ± 0.3 |
| MG | Microparticles GmbH | PS-Forschungspartikel | None | None | 3.03 | 2.97 ± 0.11 | −12.6 ± 0.3 |
| KI | Kisker Biotech GmbH & Co.KG | Polystyrene microparticles | PPS-3.0 | None | 3 | 2.96 ± 0.03 | −5.3 ± 0.5 |
| ST | Spherotech, Inc. | none | PP-30-10 | None | 3.43 | 3.47 ± 0.26 | −93.1 ± 1.1 |
| TS | ThermoFisher Scientific, Inc. | Latex Microspheres | 5300 A | None | 2.8 | 2.84 ± 0.03 | −13.7 ± 0.2 |
| TJ | Tianjin BaseLine Chromatographic Technology | Unibead PS-Microspheres | 6-1-0300 | None | 3.0 | 3.36 ± 0.07 | −45.5 ± 0.7 |
| PX | Phosphorex, Inc. | Polyspherex | 11 | None | 3.246 | 3.13 ± 0.33 | −7.5 ± 0.4 |
| MM-SW2 | Micromod GmbH | micromer® | 01-00-303 | Salt water incubation, 2 weeks | 3 | 2.97 ± 0.17 | −10.0 ± 0.7 |
| MM-SW4 | Micromod GmbH | micromer® | 01-00-303 | Salt water incubation, 4 weeks | 3 | 3.03 ± 0.14 | −7.2 ± 1.3 |
| MM-FW2 | Micromod GmbH | micromer® | 01-00-303 | Freshwater incubation, 2 weeks | 3 | 2.88 ± 0.28 | -9.2 ± 0.5 |
| MM-FW4 | Micromod GmbH | micromer® | 01-00-303 | Freshwater incubation, 4 weeks | 3 | 2.96 ± 0.04 | −16.0 ± 2.3 |

Measured diameters were determined by scanning electron microscopy (see Supplementary Note 1, Supplementary Table 1). Values of measured diameter and ζ-potential represent mean ± standard deviation. For the measured diameter, $n = 10$ particles were analyzed per sample. The ζ-potential measurements were replicated $n = 3$ times.

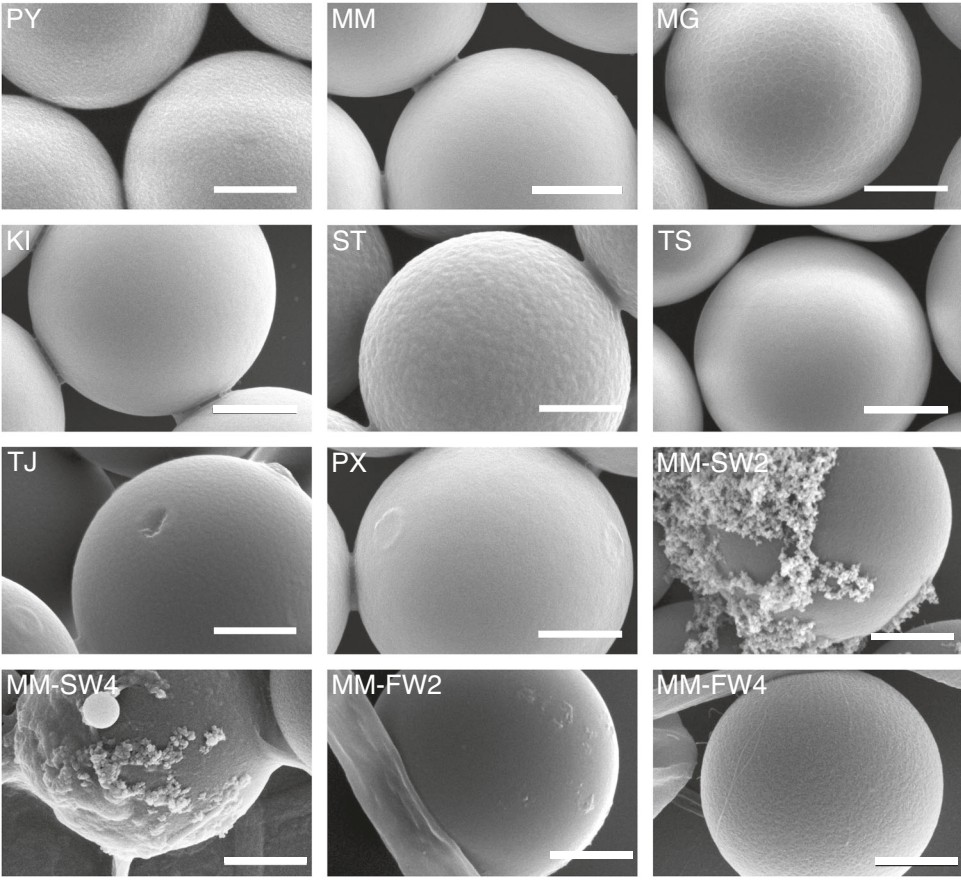

**Fig. 1 | Scanning electron microscopy micrographs of the polystyrene particles.** An overview of the abbreviations and specifications is given in Table 1. The surface morphologies of the different types of particles varied strongly. TJ had the roughest surface (see Supplementary Table 1), where TJ showed elevations and indentations and PX elevations. MG seemed to be covered by a net-like structure. All other particles are highly spherical with smooth surfaces. Those particles exposed to salt water particles (MM-SW2 and MM−SW4) have larger elevations, probably originating from salts whereas particles exposed to freshwater (MM-FW2 and MM−FW4) show rather smooth surfaces with little elevations. Scale bars: 1 µm.

## Eco-coronas affect the ζ-potential of microplastic particles

The exposure of MM microplastic particles to environmental salt and freshwater for 2 and 4 weeks lead to the formation of an eco-corona. In the environmental media, several microorganisms were present, including cyanobacteria, green algae of the genus *Lagerheimia*, and diatoms (Supplementary Fig. 3). Scanning electron microscopy showed that some of the microplastic particles were visibly coated with an eco-corona, probably originating from biomolecules released by the microorganisms (Fig. 1). Accordingly, the equivalent diameters of the particles slightly increased for MM-SW2, MM-SW4, and MM-FW4. Their respective standard deviations substantially increased for all particle types, indicating that they became less monodisperse (Table 1). Furthermore, the eccentricity and surface roughness increased for all environmentally exposed particles compared to the pristine MM particles (Supplementary Table 1).

In a previous study with identically prepared MM particles, we showed that this eco-corona forms heterogeneous polymer structures on the particles' surface with properties of anchored high molecular weight polymer coatings[52]. Furthermore, we previously identified amino acids, nucleic acids, and lipids on MM particles incubated in freshwater using Raman spectroscopy[15]. To further analyze the eco-corona in this study, we performed synchrotron-based scanning transmission X-ray microscopy (STXM, Supplementary Fig. 4, Supplementary Table 2). We observed small amounts of protein-associated C-O and sugar-associated C-OH groups on the MM particles. After incubation in freshwater, these signals significantly increased, indicating the formation of an eco-corona. However, we could not observe an increase in the amount of proteins and sugars on MM particles exposed to salt water. A potential explanation could be that parts of the eco-corona were washed off due to the change in ionic strength in the seawater-incubated sample during the rinsing procedure that was required to avoid salt precipitation during drying of the samples. As the surface sensitivity of STXM as a transmission technique is limited, we additionally used X-ray photoelectron spectroscopy (XPS, Supplementary Table 3). The XPS spectra showed that exposure of MM to salt and freshwater altered the microplastic particles' surface. We detected nitrogen on the surfaces of the environmentally exposed particles, which was absent in the pristine MM particles, possibly indicating the presence of biomolecules or other natural organic matter. Small changes in silicon and oxygen signals could not be reliably separated from potential influences of the substrate (thermally oxidized silicon wafer). On the surface of the microplastic from salt water we additionally identified traces of sodium, magnesium, sulfur, and chlorine compared to the pristine particles.

The particles exposed to environmental media had a more negative ζ-potential compared to the pristine MM particles. Like the pristine microplastic particles, the environmentally exposed microplastic particles slightly changed the magnitude of their ζ-potential after incubation in cell culture media (Supplementary Table 1, Supplementary Fig. 1).

## Particle-cell adhesion depends on the ζ-potential

We developed a microfluidic platform and used a convolutional neural network to quantify the influence of the ζ-potential on the particle-cell binding kinetics and the adhesion strength to cells. Particles were diluted to a concentration of approximately $10^7$ particles per mL in imaging medium and carefully flushed into the microfluidic channels containing the cells. In a first step, we used the microfluidic platform to analyze the diffusive motion of individual particles during the sedimentation onto the cells. By classifying binding and unbinding events from and to cells using a convolutional neural network (Fig. 2a), we quantified the average binding kinetics of each particle type with their respective on-rates $k_{on}$ and their off-rate $k_{off}$ (Fig. 2b). A high $k_{on}$ and a low $k_{off}$ corresponds to fast binding and slow unbinding respectively and therefore to a strong adhesion. Some particles bound and never detached until the end of the

experiment. We classified the corresponding binding events to be irreversible. In a second step, we exerted a tunable hydrodynamic shear force on the particles and quantified the number of remaining particles after 30 s. Using a lattice Boltzmann method, we related the flow rate in the microchannels to the hydrodynamic shear force on the particles. With this method, we therefore quantified four parameters (on-rate, off-rate, percentage of irreversible binding events and percentage of bound particles under shear force) which enable us to assess the strength of particle-cell interactions. Since different endocytic pathways such as phagocytosis depend on particle-cell binding and adhesion this binding strength is expected to be a relevant parameter for the absolute internalization probability.

The binding kinetics of different particle types varied significantly (Supplementary Table 4, Supplementary Data 1) by multiple orders of magnitude. Between particles and cells, $k_{on}$ varied between $(8.1 \pm 0.8) \times 10^{-4}\,s^{-1}$ (MM) and $2.5 \times 10^{-2}\,s^{-1}$ (ST) (Fig. 2c, Kruskal–Wallis test: two-sided $P = 1.75 \times 10^{-14}$) while $k_{off}$ varied between $(1.5 \pm 0.1) \times 10^{-4}\,s^{-1}$ (PY) and $2.5 \times 10^{-2}\,s^{-1}$ (MM) (Fig. 2d, Kruskal–Wallis test: two-sided $P = 3.00 \times 10^{-10}$). Between particles and coverslips, we measured rates of a similar magnitude (Supplementary Fig. 5). This means that particles which strongly bound to cells also bound strongly to coverslips. However, particle-coverslip adhesion was in general slightly weaker than particle-cell adhesion. This was reflected by a generally lower $k_{on}$ to coverslips and a higher $k_{off}$ from coverslips (Supplementary Fig. 5). The fraction of irreversible binding events to cells varied between $(32 \pm 11)$ % (MM) and $(98.6 \pm 0.3)$ % (PY, Fig. 2e, Kruskal–Wallis test: two-sided $P = 6.71 \times 10^{-11}$) and in a similar range for coverslips (Supplementary Fig. 5).

Previously, we showed that exposure to environmental media alters the cellular interactions and internalization of microplastic particles[15]. Therefore, we wanted to investigate whether environmental exposure affects their binding kinetics and adhesion strength to cells. We found that MM particles coated with an eco-corona, regardless of the eco-corona origin (salt or freshwater), adhered stronger to cells and coverslips than MM particles without an eco-corona. While unmodified MM particles rarely bound to cells and coverslips, particles with an eco-corona commonly bound to cells and coverslips. For example, $k_{on}$ to cells increased about an order of magnitude from $(8.1 \pm 0.8) \times 10^{-4}\,s^{-1}$ to $(8.0 \pm 0.9) \times 10^{-3}\,s^{-1}$ after two weeks in salt water and to $(8.1 \pm 1.0) \times 10^{-3}\,s^{-1}$ after two weeks in freshwater. $k_{off}$ decreased from $(2.5 \pm 0.2) \times 10^{-2}\,s^{-1}$ to $(1.4 \pm 0.1) \times 10^{-2}\,s^{-1}$ after two weeks in salt water and to $(1.1 \pm 0.1) \times 10^{-2}\,s^{-1}$ after two weeks in freshwater (Fig. 2). The fraction of irreversible binding events changed from $(32 \pm 11)$ % (MM) to $(44 \pm 6)$ % and $(46 \pm 3)$ % after two- and four-weeks exposure to salt water, and $(81 \pm 3)$ % and $(32 \pm 5)$ % after two and four weeks exposure to freshwater. We observed a similar increase of $k_{on}$, decrease of $k_{off}$, and an increase of the fraction of irreversible binding events for the interaction of eco-corona-coated MM with the coverslips (Supplementary Fig. 5).

The particle-cell as well as the particle-coverslip binding was strongly correlated to the ζ-potential (Fig. 2c–e, Supplementary Fig. 5). With increasing negative ζ-potential $k_{on}$ increased (Pearson's $R = 0.9$, two-sided $P = 4 \times 10^{-5}$), $k_{off}$ decreased (Pearson's $R = -0.9$, two-sided $P = 0.0003$), and the fraction of irreversible binding events increased (Pearson's $R = 0.8$, two-sided $P = 0.0007$). Overall, the analysis of the microplastic particle binding kinetics indicates that particles with a more negative ζ-potential interact stronger with cells than more neutral microplastic particles.

Since different interaction processes such as phagocytosis depend on the adhesion between microplastic particles and cells, we also quantified the adhesive strength under a well-defined shear force. Therefore, we exerted a constant hydrodynamic drag force of $(50 \pm 5)$ pN (Eq. (7)) for 30 s on the particles after the sedimentation phase (Fig. 3a) and determined the fraction of remaining particles, which were not ruptured off (Fig. 3b, c). The measured adhesion strengths varied

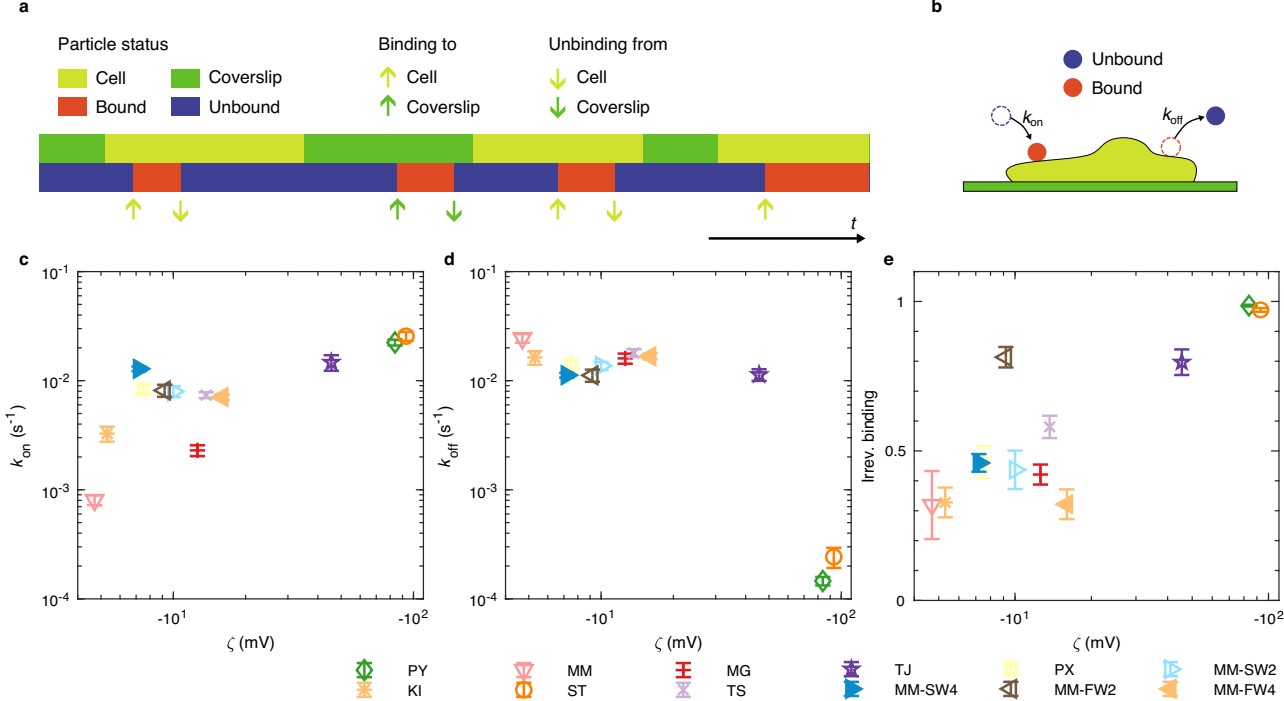

**Fig. 2 | Particle binding and unbinding kinetics. a** Schematic representation of the status of a single particle as a function of time. The light green and green bars indicate whether the particle is close to a cell (light green) or close to the coverslip (green) and the blue and red bars indicate whether the particle is bound (to either cell or coverslip, red) or unbound (blue) at a given time point. In this example, there are three binding events to a cell (light green up arrows) and two unbinding events from a cell (light green down arrows) as well as one binding event to the coverslip (green up arrow) and one unbinding event from the coverslip (green down arrow). The binding kinetics is characterized by the respective binding and unbinding rates $k_{on}$ and $k_{off}$ (**b**). **c** Binding rates $k_{on}$ to cells significantly differed between samples (Kruskal-Wallis test, two-sided $P = 1.75 \times 10^{-14}$). **d** Unbinding rates $k_{off}$ from cells significantly differed between samples (Kruskal–Wallis test, two-sided $P = 3.00 \times 10^{-10}$). **e** Fraction of irreversible binding events to cells significantly

differed between samples (Kruskal–Wallis test, two-sided $P = 6.71 \times 10^{-11}$). In general, $k_{on}$ was higher for particles with a more negative $\zeta$-potential (Pearson's $R = 0.9$), two-sided $P = 4 \times 10^{-5}$, (**c**). $k_{off}$ was lower for particles with a more negative $\zeta$-potential (Pearson's $R = -0.9$), two-sided $P = 0.0003$, (**d**). The fraction of irreversible binding events also strongly depended on the particle type. Particle-cell and particle-coverslip binding events with PY and ST particles were almost always irreversible, while 25–75% of the bonds did rupture spontaneously for MM, MG, KI, TS, MM-SW2, MM-SW4, and MM-FW4 particles. We found that the fraction of irreversible binding events is higher for particles with a more negative $\zeta$-potential (Pearson's $R = 0.8$), two-sided $P = 0.0007$, (**e**). In all panels, error bars represent standard error of mean of $n = 9$ measurements (for each measurement, on average 550 particles were analyzed). For particle abbreviations and characteristics see Table 1. Source data are provided as a Source Data file.

significantly (Supplementary Table 4, Supplementary Data 1) between particles of different suppliers. For the adhesion to cells, we observed a remaining fraction between $(3 \pm 1)$ % (MM), indicating that most of these particles were readily flushed away, and $(102 \pm 1)$ % (PY), showing that these particles were not ruptured off the cells (Fig. 3d, Kruskal–Wallis test: two-sided $P = 2.05 \times 10^{-9}$). The results for the adhesion to coverslips was similar. However generally, the adhesion strength was slightly lower in this case (Supplementary Fig. 6). Particles that strongly adhered to cells also strongly adhered to coverslips and vice versa.

The fraction of remaining particles increased for microplastic particles coated with an eco-corona (MM-SW2, MM-SW4, MM-FW2, and MM-FW4), compared to the respective particles without an eco-corona (MM). For example, the fraction of particles remaining on cells increased from $(3 \pm 1)$ % to $(24 \pm 6)$ % after two weeks in salt water and to $(18 \pm 3)$ % after two weeks in freshwater. After four weeks in salt water, the fraction of remaining particles increased to $(28 \pm 2)$ %, and after four weeks in freshwater, the fraction of remaining particles increased to $(20 \pm 1)$ %. We observed a similar increase of the fraction remaining particles on coverslips after exposure to salt and fresh water for two and four weeks (Supplementary Figure 6).

The particle-cell and particle-coverslip adhesion was strongly correlated to the microplastic particles' $\zeta$-potential (Fig. 3d, Supplementary Fig. 6). With increasingly negative $\zeta$-potential, the fraction of remaining particles on cells strongly increased (Pearson's $R = 0.95$, two-sided $P = 1.4 \times 10^{-6}$). Overall, the analysis of the number of particles remaining bound to cells even under a hydrodynamic shear force

indicate that the adhesive forces between microplastic particles and cells increase with a more negative $\zeta$-potential, while more neutral particles barely adhered to the cells.

## Absolute internalization probability depends on $\zeta$-potential

To investigate whether adhesion is a key determinant for particle internalization, we studied whether particles which adhered stronger to cells had a higher internalization probability. To this end, microplastic particles were added to the cells, which were then incubated 1 h on ice, so that the particles could sediment. Once the particles sedimented, the cells were incubated for 2 h at 37 °C so that they could internalize the microplastic particles. They were then fixed and analyzed using confocal fluorescence microscopy to quantify the number of internalized particles. The conditional internalization probability (Fig. 4a) denotes the probability that a particle is internalized by a cell if it is already attached to the cell. The highest conditional internalization probability was found for MG $(77 \pm 2)$ % particles, whereas TS particles had the lowest conditional internalization probability $(13 \pm 2)$ %, (Fig. 4a). Particles from the other manufacturers ranged between $(27 \pm 1)$ % (PY) and $(55 \pm 3)$ % (TJ). Although the microfluidics experiments showed a correlation between the $\zeta$-potential and the adhesion of the particles to cells and coverslips, the conditional internalization probability did not correlate with the $\zeta$-potential (Pearson's $R = -0.2$, $P = 0.6$). However, the probability that a microplastic particle is internalized by a cell depends on both, the probability to adhere to a cell (i.e., the adhesion strength) and the subsequent probability to get

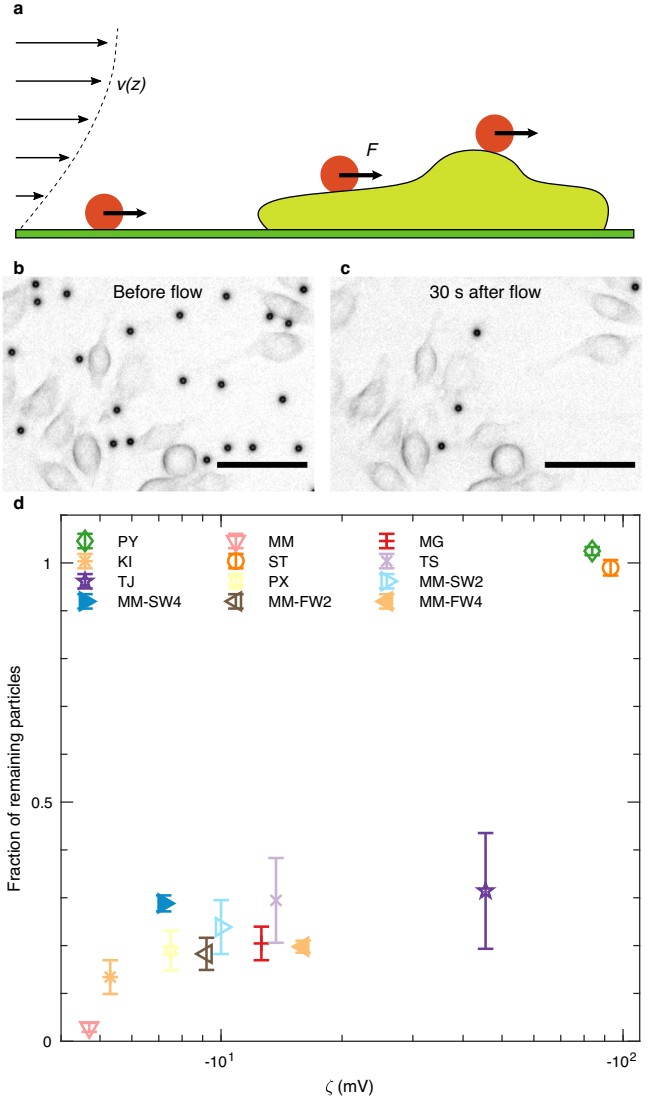

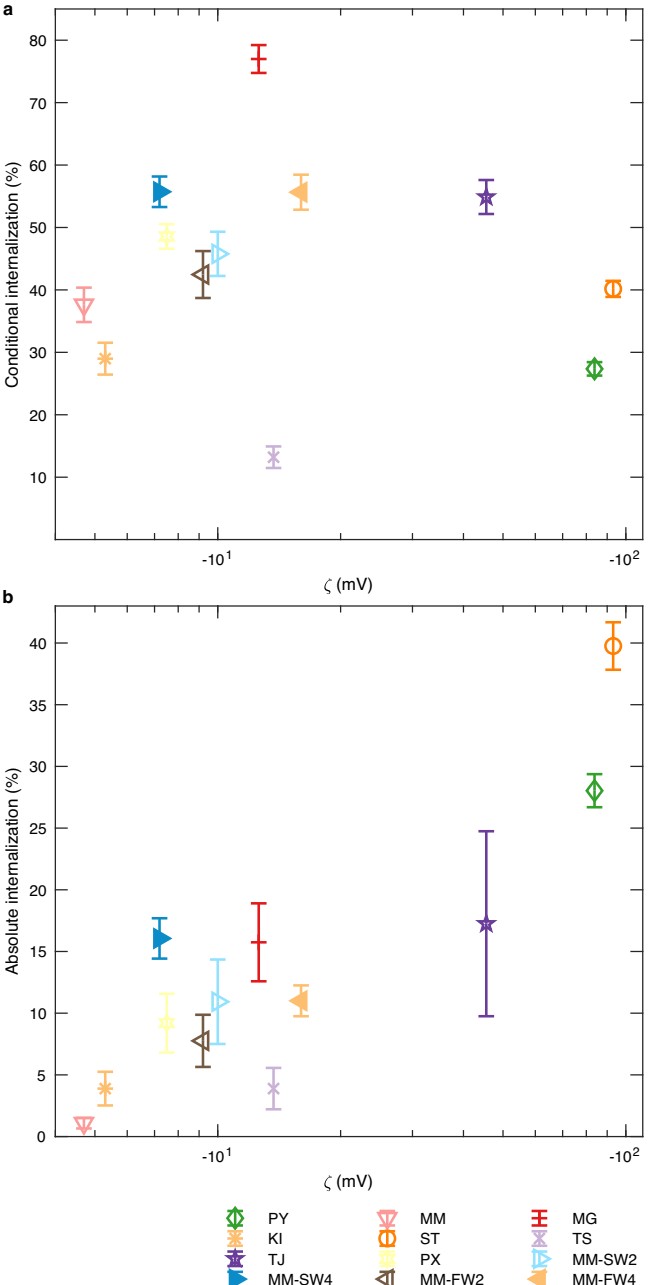

**Fig. 3 | Adhesion of microplastic particles to cells under shear force. a** After the particles sedimented and bound (red) to the cells (light green) and the coverslip (green) for 10 min in the microfluidic chamber, we turned on a Poiseuille flow with a profile $v(z)$ given by Eq. (15) in our channels, imposing a hydrodynamic force of $F = (50 \pm 5)$ pN (Eq. (7)) on the particles. **b, c** Right before the flow, all particles sedimented and attached to cells or coverslips. We determined the fraction remaining particles after 30 s of flushing. Scale bars: 50 µm. **d** Fraction of particles remaining on cells. The observed fraction of particles remaining on cells significantly differed between samples (Kruskal–Wallis test, two-sided $P = 2.05 \times 10^{-9}$). The adhesion strength of microplastic particles was strongly correlated with their $\zeta$-potential (Pearson's $R = 0.95$, two-sided $P = 1.4 \times 10^{-6}$). While neutral particles barely adhered to cells, the adhesive forces between microplastic particles and cells increased with a more negative $\zeta$-potential. In all panels, error bars represent standard error of mean of $n = 9$ measurements (for each measurement, on average 550 particles were analyzed). Source data are provided as a Source Data file.

**Fig. 4 | Conditional and absolute internalization probability of microplastic particles into cells. a** The conditional internalization probability displayed as a function of the $\zeta$-potential of the particles indicates no correlation between the two parameters (Pearson's $R = -0.2$, two-sided $P = 0.6$), whereas the absolute internalization probability (**b**) does (Pearson's $R = 0.9$, two-sided $P = 5.4 \times 10^{-5}$). The internalization probability of particles coated with an eco-corona (MM-SW2, MM-SW4, MM-FW2 and MM-FW4) was calculated from the data published by Ramsperger et al.[15]. In (**a**), error bars represent standard error of mean of $n = 3$ replicates (for each replicate, 100 particle-cell interactions were analyzed). The error bars in (**b**) were propagated from the uncertainties in (**a**) and Fig. 3d. Source data are provided as a Source Data file.

internalized if it is bound. Therefore, we determined the absolute internalization probability by multiplying the conditional internalization probability with the fraction of remaining particles (Fig. 4b). The absolute internalization probability varied by almost two orders of magnitude and ranged from $(1.1 \pm 0.4)$ % (MM) to $(40 \pm 2)$ % (ST). Furthermore, it correlated with the $\zeta$-potential (Pearson's $R = 0.9$, $P = 5.4 \times 10^{-5}$). The MM particles coated with an eco-corona showed a higher absolute internalization probability (MM-SW2: $(11 \pm 3)$ %, MM-SW4: $(16 \pm 2)$ %, MM-FW2: $(8 \pm 2)$ % and MM-FW4: $(11 \pm 1)$ %) than the unmodified MM particles without an eco-corona $(1.1 \pm 0.4)$ %, which

correlated with the more negative $\zeta$-potential of the environmentally exposed particles (Table 1).

**Microplastics internalized via actin-dependent pathways**

To verify our results about the internalization of the particles, we performed additional experiments where we investigated the internalization process using live cell imaging. Due to their size, we

expected that the microplastic particles were internalized either by phagocytosis or macropinocytosis. Both internalization mechanisms require remodeling of the actin cytoskeleton, and once the particles are internalized, they undergo a similar maturation process where they interact with lysosomes and become acidified[53–55]. To test this hypothesis, we monitored the actin cytoskeleton during particle internalization by cells that were stably transfected with a LifeAct-GFP construct. Furthermore, we treated the cells with LysoTracker dye, to monitor the subsequent maturation process (Supplementary Fig. 7). We found that all particle types undergo a similar form of internalization and maturation (Supplementary Fig. 8): First, there was a substantial peak in the LifeAct signal around the microplastic particles, indicating that filamentous actin was polymerized. Eventually, the LifeAct signal around the particles decayed, indicating depolymerization of the actin filaments and successful internalization of the microplastic particle. Subsequently, the LysoTracker signal gradually increased over time, showing that internalized microplastic particles underwent a maturation process during which they interacted with lysosomes and were acidified. We found that 100% of the particles that were acidified during the measurement time showed a LifeAct peak before the acidification process started, independent of the microplastic particle type (Supplementary Fig. 9). Overall, these results show that all particle types were internalized via phagocytosis or macropinocytosis.

## Discussion

We showed in a systematic approach that the $\zeta$-potential of nominally identical model microplastic particles differed by up to more than one order of magnitude, leading to significant differences in their particle-cell interactions. These differences were likely related to the manufacturing process of the particles, since different methods of polymerization can lead to different functional groups on the surface of the particles, originating from different surfactants, initiators, or catalyzers used[32,56–58]. These differences in microparticle properties are likely not only relevant for particles from different manufacturers, but also for different batches of the same particle type from the same manufacturer. Therefore, it is important to always thoroughly characterize the model microplastic particles that are used.

In previous studies, it has been established that the $\zeta$-potential affects the interactions of nano- and microparticles with cells[6,32,35–37]. However, especially for microparticles, the results were not unanimous. For example, with increasingly negative $\zeta$-potential, both increasing[43,44] and decreasing[41,42] interactions of microparticles and cells have been reported. We provide deeper insight here, since we systematically analyzed the cellular interactions of twelve different microparticle types spanning a wide range of $\zeta$-potentials from $-4.7$ mV to $-93.1$ mV., We individually assessed the role of the $\zeta$-potential for the microparticle binding kinetics, adhesion strength, conditional internalization probability, and absolute internalization probability.

Using our multiplexed single-cell single-particle microfluidic platform, we quantified the binding kinetics during the sedimentation of the particles, and the particles remaining attached upon exertion of 50 pN hydrodynamic force. Our results indicate that polystyrene microplastic particles of the same size and shape with a more negative $\zeta$-potential bound faster, unbound slower, and adhered stronger to the cells. Overall, these measurements agree with previous studies on the adhesive forces of different particle types. Using magnetic tweezers Chen et al.[59] reported adhesion forces of about 15 pN between the particles and coverslips for polystyrene microplastic particles coated with extracellular polymeric substances (EPS). Using much larger 10 μm sized particles, Liu et al.[42] quantified adhesive forces by atomic force microscopy (AFM) between PLA, PLGA, and PELA microparticles and cells. These measurements yielded adhesion forces of 1.63 nN (PLGA), 1.85 nN (PELA), and 2.38 nN (PLA). These forces were higher

than in our study, likely because a much larger particle size that was used in that study (11 times larger surface area than our particles) and due to the measurement method applied. With our approach, the microplastic particles sedimented freely onto cells and coverslips, whereas in AFM measurements, they were pushed with a force of several nN onto the cells, potentially leading to a larger contact area and therefore stronger interactions between particles and the cell membrane[60]. Furthermore, the forces were oriented differently in both experiments. Whereas with the AFM the forces were exerted perpendicular to the cell surface, they were oriented parallel to the surface in our approach.

In general, electrostatic interactions quantified by the $\zeta$-potential or the specific binding of ligands to membrane receptors can mediate particle-cell binding and adhesion[61]. In our experiments, we observed similar particle-cell and particle-coverslip adhesion strengths. The fact that the particles' binding strength both to cells and coverslips was strongly correlated with their $\zeta$-potential indicates that local electrostatic interactions between the charged groups on microplastic particles and the cell membrane (presumably supported by multivalent ions from the medium[43]) are highly important for their binding strength.

This might be also true for the coating with biomolecules from salt and freshwater forming an eco-corona on the surface of a microplastic particle. We showed in previous works that environmental exposure substantially alters the surface of microplastic particles and leads to the formation of an eco-corona[15,52]. In this study, we additionally performed SEM imaging, STXM, and XPS to further quantify the eco-corona structure and constituents. The constituents creating an eco-corona on the surface of a microplastic particle were previously described as proteins, humic and fulvic acids, amino acids, lipids, polysaccharides, and carbohydrates[15,62–64]. Consistently, the STXM measurements showed an increase in proteins and sugars on the surface of freshwater-exposed microplastic particles, and the XPS measurements indicated the presence of organic nitrogen on salt and freshwater-exposed microplastic particles. Molecules like humic and fulvic acids have multiple carboxylic groups, carrying negative charges[65–67]. As these make up the largest fraction of natural organic matter[68], they potentially caused the more negative $\zeta$-potential of the environmentally exposed microplastic particles in our study. Therefore, different charged sites and different densities of the charged sites on the surface of the particles lead to different electrostatic forces between particles and cellular membranes and between particles and coverslips.

This is in concordance with earlier works that already indicate that the adhesion of a particle to cellular membranes is mainly driven by the surface charge of a particle[43,69,70]. However, the cellular membrane overall has a negative surface charge, which led to the widely accepted assumption that the binding of positively charged particles to cellular membranes is more likely than the binding of negatively charged particles[69,71,72]. Nevertheless, there already was evidence that also negatively charged particles can bind to cellular membranes[43,71,72]. The binding of negatively charged particles to on average negatively charged cell membranes is potentially supported by a heterogeneous surface charge of cells. Perry et al.[39] demonstrated that the surface of human adipocytes is locally positively charged, with μm-sized patches with charge densities of up to 50 mC m$^{-2}$, among areas with an average negative charge density of $-15$ mC m$^{-2}$. Furthermore, in the presence of multivalent positive ions, overcharging of the negative surfaces can occur[73,74]. Overcharging describes the process, when a multivalent positive ion is attracted by a monovalent negative surface group. This leads to a local overcompensation of the negative surface charge. This mechanism is also involved in the Schulze-Hardy rule that describes the limits of colloidal stability in solutions of multivalent ions[75–77]. Therefore, particles with a more negative $\zeta$-potential may interact stronger with the cells. However, some microplastic particle types,

which had similar ζ-potentials, differed slightly in their binding kinetics and adhesion. These differences may reflect ligand-receptor interactions, for example due to binding of the polystyrene or surfactants to scavenger receptors[78].

Furthermore, in the environment, microplastics are exposed to numerous natural factors such as UV irradiation[79] and eco-corona formation[15,80] which can change its ζ-potential and therefore its interactions with cells. Consistently, we observed that the eco-corona-coated microplastic particles had a more negative ζ-potential compared to the uncoated particles, and therefore also a higher binding strength. Additionally, the higher mobility of the charged groups in the eco-corona may facilitate an optimal arrangement of the charged groups on the incubated particle and the coverslip or cells. The higher portion of optimally aligned charged groups can therefore increase the adhesion force and subsequent internalization.

We showed that the microplastic particles were internalized either by phagocytosis or macropinocytosis. For phagocytosis, the adhesion of microplastic particles to cells is a prerequisite for their internalization. In the case of macropinocytosis, both particles that adhered to the cell membrane and freely diffusing particles can be internalized. However, the probability of macropinocytosis is higher for particles adhered to the cell surface since their residence time in the direct vicinity of the cell is increased compared to freely diffusing particles. Therefore, the internalization of microplastic particles can be regarded as a two-stage process[38]. First, the microplastic particles adhere to the cell membrane. Second, the attached particles are internalized. Here, we demonstrated that microplastic particle-cell adhesion is strongly determined by the ζ-potential of the particles. However, the conditional internalization probability of the different microplastic particles did not correlate with the ζ-potential. This may indicate that for the process of internalization, unspecific electrostatic forces play a minor role, whereas the biological identity of a particle plays a larger role for the internalization.

Depending on the particles' surface groups, macrophages can internalize for example polystyrene, silicon, and metal microparticles via scavenger receptor-mediated phagocytosis[78,81,82]. Unlike phagocytosis, which is tightly controlled by these receptor-ligand interactions, macropinocytosis is a more stochastic process. Nevertheless, it is not completely receptor-independent, as some receptors like EGFR can enhance the formation of membrane ruffles, which lead to increased rates of macropinocytosis[83,84]. Therefore, different chemical surface groups of the microplastic particles in our study might affect their interactions with different macrophage receptors, leading to varying conditional internalization probabilities. However, since adhesion facilitates particle internalization, the absolute internalization probability was strongly correlated with the ζ-potential.

Furthermore, biomolecules present in the eco-corona of microplastic particles may enhance internalization. Such biomolecules have been reported to trigger endocytic pathways such as scavenger-receptor mediated phagocytosis, increasing the overall conditional internalization probability[15,85,86]. Consistently, we observed an increased conditional internalization probability for the eco-corona particles (MM-FW2, MM-FW4, MM-SW2, and MM-SW4) compared to pristine particles (MM) without an eco-corona. Additionally, internalization of environmentally exposed microplastic particles is enhanced due to the increased particle-cell binding affinity.

Because of this two-step internalization process, particle-cell adhesion is a very important part for particle internalization. It has been reported that adhesion and internalization are prerequisites for cytotoxicity[32]. Thus, at similar concentrations, particles with a more negative ζ-potential can potentially be more toxic than particles with a ζ-potential close to 0, since interactions with cells are more likely[32]. To ensure that ecotoxicological studies on microplastics are comparable with each other, it will be of utmost importance to thoroughly characterize the model microplastic particles that are used, because even

nominally identical particles can strongly differ in their ζ-potential. We want to emphasize that this is also relevant for experiments using the same particle type from the same manufacturer, as batch-to-batch variations may occur. This will be equally relevant for effect studies using other polymer types, as these can similarly differ in their ζ-potential[80]. Furthermore, the ζ-potential of microplastic particles can additionally be modified by the adsorbed biomolecules forming an eco-corona. Therefore, the environmental exposure in complex eco-systems likely affects the hazard potential of the microplastic particles, making further studies in this direction necessary.

With our study we highlight that nominally identical particles from various manufacturers differ in their ζ-potential and in their interactions with cells. We identified the ζ-potential as one of the major drivers for particle-cell adhesion and consequently the absolute internalization probability. We also demonstrated that environmental exposure of microplastic particles alters their ζ-potential and thus their internalization probability as well. With our microfluidic approach, we enable an efficient quantification of the binding kinetics and adhesion strength of single particles attached to single cells in a highly multi-plexed manner. Due to the importance of the ζ-potential for the absolute internalization probability, the choice of model microplastics may drastically impact the results of microplastic effect studies, since cellular interactions and internalization of microplastic particles are one prerequisite for their toxicity[32,87]. As the ζ-potential of microplastic particles additionally changes with the formation of an eco-corona, the environmental exposure in complex ecosystems likely affects the hazard potential of microplastic particles.

## Methods
### Microplastic particles
Polystyrene particles were purchased from the following different manufacturers: Polysciences, Inc. (Warrington, PA), Micromod Partikeltechnologie GmbH (Rostock, Germany), Microparticles GmbH (Berlin, Germany), Kisker Biotech GmbH & Co.KG (Steinfurt, Germany), Spherotech Inc. (Lake Forest, IL), ThermoFisher Scientific Inc. (Waltham, MA), Tianjin BaseLine Chromatographic Technology (Tianjin, China) and Phosphorex, Inc. (Hopkinton, MA) (see Table 1). Particles obtained from Tianjin BaseLine Chromatographic Technology were provided as a powder, whereas all other particles were provided in an aqueous solution. The particles from Tianjin BaseLine Chromtech were dispersed in ultrapure water.

### Environmental exposure
Microplastic particles from Micromod Partikeltechnologie GmbH (MM, see Table 1) were exposed to environmental media as described by Ramsperger et al.[15] 100 μL of the particle stock solution was dispersed in 900 μL environmental media (salt water from a sea water aquarium and freshwater from an outside freshwater pond) in a 1.5 mL autosampler vial. To prevent sedimentation of the particles, the vials were placed on a sample roller. To keep the microbial communities in the respective media intact, the salt water and freshwater was replaced 3× per week. For that, samples were centrifuged for 20 min at 2000× *g*, 900 μL of the supernatant was discarded and replaced with 900 μL of fresh environmental media. Microplastic particles were incubated for 2 and 4 weeks, respectively.

### Microplastic particle characterization
**Scanning electron microscopy.** To investigate the surface structures of microplastic particles, samples were analyzed using a scanning electron microscope (SEM, FEI Apreo Volumescope, Thermo Fisher Scientific, 5 kV, working distance 10 mm, Everhart-Thornley detector for qualitative images; 3 kV, WD 5 mm, T1 in-lens detector for quantitative analysis). First, each stock solution of the microplastic particles was diluted in ultrapure water (1:100), and 100 μL of this dilution was pipetted onto a silicon wafer placed on carbon conductive tabs (Ø

12 mm Plano GmbH, Wetzlar, Germany) fixed to aluminum stubs (Ø 12 mm, Plano GmbH, Wetzlar, Germany). To preserve the eco-corona, the environmentally exposed beads were fixed using Karnovsky's fixative (2% PFA (reagent grade, Sigma Aldrich, Merck KgA, Germany) and 2.5% glutaraldehyde (for electron microscopy, Carl Roth GmbH, Germany) in 1× PBS) prior to dehydration in an ethanol series (30%, 50%, 70%, 80%, 90% for 30 min each, 95% and absolute ethanol for 1 h each, Ethanol purity >99.9%, VWR International S.A.S., France) and dried in hexamethyldisilazane (HMDS, purity > 98%, Carl Roth GmbH, Germany)[88]. The stubs were then transferred into a desiccator and stored until the images were acquired. Samples were subsequently coated with a 4 nm-thick platinum layer (208HR sputter coater, Cressington, Watford, UK) and analyzed using the SEM.

SEM micrographs (pixel size: 2.08 nm) were analyzed quantitatively using a custom-coded SEMParticleAnalyzer MATLAB program. The micrographs were filtered using a median filter with a radius of 3 pixels. To automatically detect the microplastic particles in the micrographs, the gradient of the images was calculated. This gradient image was binarized by choosing a suitable threshold, usually around 30–35% of the maximal pixel value. Then, binary components were dilated by 5 pixels, and remaining holes were filled. The resulting binary components were eroded by 5 pixels. Components smaller than $5 \times 10^5$ pixels (equivalent diameter of $1\,\mu m$) and components touching the border of the image were discarded.

To analyze the particle properties, the equivalent diameter, major axis length, minor axis length, and the perimeter of the particles were evaluated using the regionprops() function of MATLAB. We analyzed the equivalent diameter, the eccentricity, and the roughness of the particles. We defined the eccentricity as the quotient of the major and minor axis length. An eccentricity value of 1 would correspond to perfectly spherical particles, larger values indicate aspherically shaped particles. The roughness was defined as the particles' perimeter divided by the perimeter of a circle with the same equivalent diameter. Roughness values of 1 indicate perfectly smooth surfaces, larger values indicate an increased surface roughness. Due to the median filtering and dilation/erosion during image segmentation, only surface irregularities on length scales larger than 10 nm were detected.

**ζ-Potential.** ζ-potential was measured with a Zetasizer Nano ZS (Malvern Panalytical, Worcestershire, UK) at 24 °C after an equilibration time of 120 s. The ζ-potential was obtained by 3 single measurements with 50 runs each lasting 1 s. Particles were dispersed in 1 mM KCl. The pH in the samples ranged from 5.5 to 6.1, and the conductivity from $0.18\,mS\,cm^{-1}$ to $0.22\,mS\,cm^{-1}$. To measure the influence of an incubation of the particles in cell culture media, the particles were incubated for 2 h in 9 mL cell culture media at a concentration in $1.5 \times 10^6$ particles $mL^{-1}$. After the incubations, particles were centrifuged for 20 min at 2000× g, washed 1× in 1 mM KCl, and resuspended in 1 mL 1 mM KCl. Then, their ζ-potential was measured as described above. The pH in the incubated samples ranged from 5.4 to 6.7, and the conductivity from $0.16\,mS\,cm^{-1}$ to $0.21\,mS\,cm^{-1}$.

**Synchrotron-based scanning transmission X-ray microscopy.** Samples for STXM analysis were gently rinsed in DI water to avoid salt precipitation during drying. Samples were wet deposited from aqueous suspensions onto formvar coated 300 mesh Cu TEM grids, blotted and dried immediately. Samples were analyzed at beamline 10ID-1 at the Canadian Light Source. Image stacks across the C1s absorption edge were recorded between 275 and 340 eV with 0.1 eV steps in the energy region of interest. Images were aligned and converted from transmission to linear absorbance scale (optical density (OD)) using aXis2000[89]. The surface regions were selected in the STXM images based on the average optical density (OD) range of 0.1–0.9 across the C1s absorption edge, which is equivalent to a cumulative

thickness of up to 100 nm, arranged tangentially around the PS particles. The equivalent thickness was calculated using the atomic scattering factors[90], the formula $C_8H_8$ and an assumed density of $1.09\,g\,cm^{-3}$ of the polystyrene. All 3 spectra were decomposed into a sum of individual gaussian peaks plus the ionization edge modelled as an arctan function. A minimum of 7 analytical peaks was required and used for fitting the respective spectra: 284.0 eV (quinone C = O), 285.0 and 285.4 eV (aromatic C = C), 287.4 eV (aliphatic C-C), 288.2 eV (protein C-O), 288.9 eV (carboxylic C-O), 289.5 (polysaccharide C-O). Peak energies and widths were optimized and fixed at the same values for all 3 spectra, whereas the respective peak areas were fitted using the peak fitting algorithm of Athena[91].

**X-ray photoelectron spectroscopy.** All XPS studies were carried out by means of an Axis Ultra photoelectron spectrometer (Kratos Analytical, Manchester, UK). The spectrometer was equipped with a monochromatic Al K-alpha (1486.6 eV) X-ray source of 300 W at 15 kV. The kinetic energy of photoelectrons was determined with hemispheric analyzer set to pass energy of 160 eV for wide-scan spectra and 20 eV for high-resolution spectra. During all measurements, electrostatic charging of the sample was avoided by means of a low-energy electron source working in combination with a magnetic immersion lens. Later, all recorded peaks were shifted by the same value that was necessary to set the component peak of the $sp^3$-hybridized carbon atoms to 285.00 eV. The polystyrene particles were deposited as a particle film from their aqueous suspension on a thermally oxidized silicon wafer. Quantitative elemental compositions were determined from peak areas using experimentally determined sensitivity factors and the spectrometer transmission function. The spectrum background was subtracted according to Shirley[92]. The high-resolution spectra were deconvolved by means of the Kratos spectra deconvolution software. Free parameters of component peaks were their binding energy (BE), height, full width at half maximum and the Gaussian-Lorentzian ratio.

**Cell lines and cell culture conditions.** Murine macrophage J774A.1 cells (DSMZ, Braunschweig, Germany) and a stable J774A.1 cell line transfected with a LifeAct-GFP construct[93] were cultured under standard culture conditions (37 °C, 5% CO_2, humidified) in Dulbeccos's Modified Eagle's Medium (DMEM, Thermo Fisher Scientific Inc., Waltham, MA), supplemented with 10% (v/v) FCS (Thermo Fisher Scientific Inc., Waltham, MA) and 1% penicillin/streptomycin (Thermo Fisher Scientific Inc., Waltham, MA)[15,94]. To maintain suitable cell concentrations, cells were passaged three times per week and cultured in T-25 culture flasks (CORNING, New York, USA).

**Microfluidics.** Custom-built flow chambers were used to quantify the adhesion strength between microplastic particles and cells. Flow chambers were built from a plastics top part (sticky-Slide I Luer, nominal channel height 0.1 mm, nominal width 5 mm, nominal length 48 mm, ibidi GmbH, Gräfelfing, Germany), which was glued to a glass coverslip (24 mm × 60 mm, #1, Menzel Gläser, Thermo Fisher Scientific Inc., Waltham, MA) with a thin film of epoxy resin one day before the experiments.

Prior to the microfluidic experiments, the cells from culture stocks were scraped off the culture flasks into the culture medium, centrifuged (150× g, 2 min, 20 °C) and re-suspended with 600 μL of cell culture medium. Into each flow chamber, 200 μL of cell suspension was added and transferred back into the incubator at 37 °C and 5% CO_2 for 1 h until the cells adhered to the bottom coverslip of the flow chamber.

Live cell imaging was performed with a frame rate of 1 Hz on an inverted, motorized microscope (Nikon Eclipse Ti, Nikon, Tokyo, Japan) with a 10× objective (CFI Plan Fluor DL 10×, Nikon, NA = 0.3), which was equipped with a CCD camera (pco.pixelfly usb, PCO AG,

Kehlheim, Germany). The microscope body was enclosed in a custom-built incubation chamber which keeps the body of the microscope and the sample at a temperature of 37 °C. A high precision linear stage (L511.20DG10, Physik Instrumente, Karlsruhe, Germany) was used to drive the piston of a syringe ($r_{piston} = 6.135$ mm) with a controlled motor velocity $v_m$ to control the flow in the chamber. The syringe was connected to the flow chamber with a tubing system, in which check valves (RVMINI-32, Piper Filter GmbH, Zwischenahn, Germany) were used similarly to the way diodes are used to build a bridge rectifier in an electrical circuit to ensure the flow direction inside the chamber stays the same after the motor reverses its direction. The motor and the camera were controlled with a custom-written MATLAB program (MATLAB 2019b, The MathWorks Inc, Natick, MA), which was used to set the motor velocity according to the channel geometry and the desired force of 50 pN exerted on the particles during the experiments. Thus, even though the geometry of different channels differed slightly, we were able to exert reproducible forces on particles in different channels.

All microfluidic experiments were performed in imaging medium (Minimum Essential Medium), (Thermo Fisher Scientific Inc.) supplemented with 5% HEPES (Thermo Fisher Scientific Inc.) as a pH buffer and 1% penicillin-streptomycin. The medium was pre-warmed to 37 °C overnight to free it from dissolved gas and avoid bubble formation during the experiments. The microparticles were added to the microfluidic system and were briefly dispersed in the microfluidic system right before the first experiment started. Furthermore, we repeated the experiments 40 min and 80 min after the particles were added to the microfluidic system. Each time series for each particle type was replicated three times, yielding 9 experiments per particle type. For each experiment, on average 550 particles were analyzed.

### Derivation of the hydrodynamic drag force on the microparticles

**Poiseuille flow in a rectangular channel.** The laminar flow in a rectangular channel with length $L$, width $w$ and height $h$ (Supplementary Fig. 10) has the velocity profile[95,96]

$$v_x(y,z) = \frac{4h^2\Delta p}{\pi^3\eta L}\sum_{n,\text{odd}}^{\infty}\frac{1}{n^3}\left[1-\frac{\cosh\left(n\pi\frac{y}{h}\right)}{\cosh\left(n\pi\frac{w}{2h}\right)}\right]\sin\left(n\pi\frac{z}{h}\right), \quad (1)$$

whereby $-\frac{w}{2} \leq y \leq \frac{w}{2}$ and $0 \leq z \leq h$ are the coordinates in the channel cross section. $\rho$ is the density of the fluid and $\eta$ is the dynamic viscosity of the fluid. $\Delta p$ is the pressure drop along the $x$-direction, which is defined by the volume flow rate $Q$[96]:

$$\Delta p = \frac{12\eta LQ}{wh^3}\left[1-\sum_{n,\text{odd}}^{\infty}\frac{192h}{n^5\pi^5 w}\tanh\left(n\pi\frac{w}{2h}\right)\right]^{-1} \quad (2)$$

**Estimation of the expected force on the particle.** If a spherical particle is located in the center of the chamber ($y = 0$), it experiences a drag force due to the laminar fluid flow. This drag force is first estimated with Stokes drag using the flow velocity at the sphere center. To approximate this velocity, the channel is assumed to have infinite width $w \gg h$, meaning the Poiseuille flow is approximated to be only two-dimensional (2D):

$$v_x(z) = \frac{\Delta pw}{2\eta L}\left(\left(\frac{h}{2}\right)^2-\left(z-\frac{h}{2}\right)^2\right). \quad (3)$$

The pressure gradient $\Delta p$ for a given volume flow rate $Q$ is then:

$$\Delta p = \frac{12\eta L\,Q}{wh^3} \quad (4)$$

In this model system, we place the spherical particle directly at the bottom wall of the channel and neglect the effect of the cell on the flow. To approximate the viscous drag force on the sphere, we evaluate the velocity at the center of the sphere at $z = r$, i.e. one sphere radius away from the bottom wall at $z = 0$, resulting in

$$v_x(z=r) = \frac{\Delta pw}{2\eta L}\left(\left(\frac{h}{2}\right)^2-\left(r-\frac{h}{2}\right)^2\right) = 6Q\frac{r(h-r)}{wh^3} \quad (5)$$

which is inserted in the Stokes drag force on a sphere ($F = 6\pi\eta rv$):

$$F \approx 36\,\pi\eta Q\frac{r^2(h-r)}{wh^3} \quad (6)$$

This approximation represents a lower estimate of the hydrodynamic force on the particle in our experiments. Firstly, the channels used in this work had a slightly parabolic height profile $h = h(y)$, which, leads to an increased flow velocity in the center of the channel compared to a rectangular channel geometry. Secondly, the presence of the channel wall modifies the force estimate from Eq. (6) which strictly holds in an infinite medium only. As the force on the particle is directly proportional to the flow velocity in the vicinity of the particle, we account for both effects by two correction factors $C_1$ and $C_2$, which will be derived in the following sections.

$$F \approx 36C_1C_2\,\pi\eta Q\frac{r^2(h-r)}{wh^3} \quad (7)$$

Equation (7) was used to calculate the required motor velocity $v_m$ to achieve a force of 50 pN using the relation

$$v_m = \frac{Q}{\pi r_{piston}^2} \quad (8)$$

**Influence of the parabolic height profile: Derivation of $C_1$.** At a given pressure drop $\Delta p$, the flow rate $Q$ in a long, rectangular channel is given by Bruus[96]:

$$Q(h,w,L) = \frac{h^3 w\Delta p}{12\eta L}\left[1-\sum_{n,\text{odd}}^{\infty}\frac{192h}{n^5\pi^5 w}\tanh\left(n\pi\frac{w}{2h}\right)\right]. \quad (9)$$

The channels that we used were not perfectly rectangular but had a slightly curved shape at the top along the $y$ axis, which could be approximated with a parabola. In this case, the channel height is given by:

$$h(y) = h_0 + \alpha y + \beta y^2. \quad (10)$$

Typically, the channels were 150 to 200 μm high in the middle and about 10 to 20 μm thinner near the side walls. To attribute for this, we calculated the corrected pressure drop in the channel in the following manner. When the boundary effects of the side walls of the channel are negligible, i.e. $h/w \to 0$ Eq. (9) can be simplified to:

$$Q \approx \frac{h^3 w\Delta p}{12\eta L}. \quad (11)$$

The total flow rate through a channel with a parabolic height profile can be approximated as the sum of flow rates through infinitesimally thin rectangular channels with varying height, considering only the boundary effects at the bottom and at the top of the infinitesimally thin channels with width d$y$:

$$Q \approx \int_{y=-w/2}^{w/2}\mathrm{d}Q = \int_{y=-w/2}^{w/2}\frac{h(y)^3\,\Delta p_{par}}{12\eta L}\,\mathrm{d}y. \quad (12)$$

Comparing Eqs. (11, 12), we can identify the effective channel height of the curved channel as:

$$h_{\mathrm{eff}} = \left( \frac{1}{w} \int_{y=-w/2}^{w/2} h(y)^3 \, \mathrm{d}y \right)^{1/3}$$
$$= \left[ h_0^3 + \frac{1}{4} h_0^2 \beta w^2 + \frac{1}{4} h_0 \alpha^2 w^2 + \frac{3}{80} h_0 \beta^2 w^4 + \frac{3}{80} \alpha^2 \beta w^4 + \frac{1}{448} \beta^3 w^6 \right]^{1/3} . \tag{13}$$

Taking the full channel profile into account, we model the pressure drop in the curved channel at a given flow rate by:

$$\Delta p_{\mathrm{par}}(h_{\mathrm{eff}}) \approx \frac{12 \eta L Q}{h_{\mathrm{eff}}^3 w} \left[ 1 - \sum_{n,\mathrm{odd}}^{\infty} \frac{192 h_{\mathrm{eff}}}{n^5 \pi^5 w} \tanh\left( n\pi \frac{w}{2 h_{\mathrm{eff}}} \right) \right]^{-1}, \tag{14}$$

and consequently, the velocity profile in a parabolic channel is approximated in analogy to Eq. (1):

$$v_{x,\mathrm{par}}(y,z) \approx \frac{4 h(y)^2 \Delta p_{\mathrm{par}}(h_{\mathrm{eff}})}{\pi^3 \eta L} \sum_{n,\mathrm{odd}}^{\infty} \frac{1}{n^3} \left[ 1 - \frac{\cosh\left( n\pi \frac{y}{h(y)} \right)}{\cosh\left( n\pi \frac{w}{2 h(y)} \right)} \right] \sin\left( n\pi \frac{z}{h(y)} \right), \tag{15}$$

This approximation corrects for the pressure change due to the parabolic height profile and ensures that the velocity at the boundary of the channel is 0. Thus, the velocity correction factor $C_1$ can be approximated by:

$$v_{x,\mathrm{par}}(y,z) \approx \frac{\Delta p_{\mathrm{par}}(h_{\mathrm{eff}})}{\Delta p(h_0)} v_x(y,z). \tag{16}$$

Since $\Delta p_{\mathrm{par}}$ was always larger than $\Delta p$ in our experiments, at the given flow rate $Q$ in the experiment, the velocity in the center $v_x(0,z)$ was always larger than it would have been if the channel had been rectangular with a height $h_0$. Since the force on the particle is proportional to the velocity in the channel, $C_1$ can be identified to be:

$$C_1 \approx \frac{\Delta p_{\mathrm{par}}(h_{\mathrm{eff}})}{\Delta p(h_0)}. \tag{17}$$

Assuming that $h_{\mathrm{eff}} \ll w$, $C_1$ simplifies to:

$$C_1 \approx \frac{h_0^3}{h_{\mathrm{eff}}^3}. \tag{18}$$

The chambers used during the experiments had a typical height in the range between 150 and 175 μm. Typically, $h_0$ was on the order of 165 μm, $h_{\mathrm{eff}}$ was about 155 μm and consequently, $C_1$ was about 1.2, indicating that the force increased by about 20%. Thus, the height profile was measured for every channel before the experiment by focusing on the top and on the bottom layer of the channels with a 40× water immersion objective and by noting the $z$ positions of the (motorized) objective. We validated the resulting velocity profile experimentally and tested whether the velocity inside the channel scales linearly with the flow rate (Supplementary Note 2, Supplementary Figs. 11, 12).

**Influence of the particle on the flow profile: Derivation of $C_2$ via lattice Boltzmann method simulations.** The correction factor $C_2$ for the force on the particle (see Eq. (7)) was determined by lattice Boltzmann method (LBM) simulations with the software FluidX3D[97]. To reduce floating-point errors and improve the overall accuracy, units were converted from SI-units to simulation units and back. To distinguish these two-unit systems, we introduced the superscripts "SI" and "sim".

A spherical microplastic particle with radius $r^{\mathrm{SI}} = 1.5$ μm was adhered to the bottom center of a rectangular microchannel with the dimensions (19.0,1.0,0.1) mm. The flow rate was $0.1$ μL s$^{-1} \leq Q^{\mathrm{SI}} \leq 50.0$ μL s$^{-1}$. For the fluid, we assumed the density and viscosity of water at $T^{\mathrm{SI}} = 37\,°\mathrm{C}$. In SI-units the given parameters were: Particle radius $r^{\mathrm{SI}} = 1.5 \times 10^{-6}$ m; channel dimensions $L^{\mathrm{SI}} = 19.0 \times 10^{-3}$ m, $w^{\mathrm{SI}} = 1.0 \times 10^{-3}$ m, $h^{\mathrm{SI}} = 0.1 \times 10^{-3}$ m; volume flow rate $Q^{\mathrm{SI}} = [0.1, 50.0] \times 10^{-9}$ m$^3$ s$^{-1}$; fluid density $\rho^{\mathrm{SI}} = 993.36$ kg m$^{-3}$; fluid dynamic viscosity $\mu^{\mathrm{SI}} = 0.6922 \times 10^{-3}$ kg m$^{-1}$ s$^{-1}$; fluid kinematic shear viscosity $\nu^{\mathrm{SI}} = \frac{\mu^{\mathrm{SI}}}{\rho^{\mathrm{SI}}} = 6.968 \times 10^{-7}$ m$^2$ s$^{-1}$. The force on the particle was then computed as the sum of the forces on all lattice points making up the particle[97] and the simulation was run until the force value converges.

The simulations were conducted with the FluidX3D software[97–99] on an AMD Radeon VII graphics processing units (GPUs) with 16 GB memory. With these memory limitations in mind, we simulated only the neighborhood of the particle and at the edge of the simulation domain. We set the velocity via moving bounce-back boundaries[97]. The simulation box dimensions were

$$L_{\mathrm{box}} = w_{\mathrm{box}} = k\, r \tag{19}$$

$$h_{\mathrm{box}} = \left( \frac{k}{2} + 1 \right) r \tag{20}$$

with $k = 16$ being chosen as large as GPU memory allows. At $z = 0$ there was a non-moving boundary representing the bottom channel wall. The other simulation box boundaries did not coincide with the channel boundaries. The particle was placed at

$$\mathbf{x}_0 = \left( \frac{L_{\mathrm{box}}}{2}, \frac{w_{\mathrm{box}}}{2}, r + 1 \right)^T \tag{21}$$

Setting the velocity at the simulation box boundaries (other than at $z = 0$ to $\mathbf{v}_x(y,z)$ (Eq. (15))) would enforce straight streamlines at the boundaries, thereby artificially constricting the flow and increasing the force on the particle (case A). However, we also could not set the boundary velocity to the analytic velocity for a laminar flow around a sphere with the $z$-dependent Poiseuille flow velocity $\mathbf{v}_x(y = 0, z)$, as this would not constrict the flow even infinitely far away from the particle, so the force would be too small (case B).

In the rectangular channel, the channel walls enforce straight streamlines. To minimize the difference of the force between the cases A and B, which confine the possible force corridor, the boundaries must be as far away as possible, but the particle still must be resolved sufficiently, so we chose $r^{\mathrm{sim}} = 16$ as a compromise for all simulations. The true force is somewhere in between the forces given by case A and case B and an interpolation of the velocities of both variants will give the best results (case C). We determined the interpolation factor as the volume fraction of the simulated volume to the total volume of the microchannel. The interpolation factors used in case C for the velocity boundaries at $r^{\mathrm{sim}} = 16$ were $\eta(v_A) = 11.39\%$ and $\eta(v_B) = 1 - \eta(v_A) = 88.61\%$.

We ran simulations for a volume flow rate of $Q^{\mathrm{SI}} \in \{0.1, 1.0, 2.0, 3.0, 4.0, 5.0, 7.0, 10.0, 15.0, 20.0, \ldots, 50.0\}$ μL s$^{-1}$ for the boundary definitions (A), (B) and (C). During each set of simulations, we kept the velocity $v^{\mathrm{sim}}$ in simulation units constant while varying the kinematic shear viscosity $\nu^{\mathrm{sim}}$ in simulation units to prevent too small $v^{\mathrm{sim}}$ and large variations in $\nu^{\mathrm{sim}}$ that would decrease the simulation accuracy. By setting $\nu^{\mathrm{sim}} = 1$ for the midway flow rate $Q^{\mathrm{SI}} = 25$ μL s$^{-1}$, we determined the corresponding velocity

$$v_x^{\mathrm{sim}} = \frac{\nu^{\mathrm{sim}} r^{\mathrm{SI}}}{\nu^{\mathrm{SI}} r^{\mathrm{sim}}} v_x^{\mathrm{SI}} \left( y^{\mathrm{SI}} = 0, z^{\mathrm{SI}} = \frac{h^{\mathrm{SI}}}{2}, w^{\mathrm{SI}}, h^{\mathrm{SI}}, Q^{\mathrm{SI}} \right) \tag{22}$$

in simulation units at the channel center for all simulations in one row. This velocity was numerically evaluated to be $v_x^{\mathrm{sim}}(y = 0, z = \frac{h^{\mathrm{sim}}}{2}) =$

0.053846 and the fluid velocity in simulation units at the center of the particle was numerically evaluated to be $v_x^{\text{sim}}(y=0, z=r^{\text{sim}})=0.003182$ for all $Q^{\text{SI}}$. When during the simulation row $Q$ was varied, the kinematic shear viscosity in simulation units varied between $\nu_x^{\text{sim}} \in [0.5, 250.0]$ for $Q^{\text{SI}} \in [0.1, 50.0]\,\mu\text{L s}^{-1}$, while $Q^{\text{sim}}$ remained constant. Our Radeon VII GPUs allowed for $k=36$ at $r^{\text{sim}}=16$ or a box size of (576, 576, 304).

The force increases linearly with the flow rate in agreement with the prediction of Eq. (7) (Supplementary Fig. 13). We fitted $F(Q)$ with Eq. (7) with $C_1=1$, to get the correction factor $C_2$:

$$C_2 = 1.618 \pm 0.002 \tag{23}$$

### Analysis of the microfluidics experiments

To quantify the transition kinetics between the bound and unbound state during the sedimentation phase, the particles were detected in every frame with a custom-written MATLAB algorithm. The detection algorithm was based on cross correlation, comparing the frames in the video with a reference image of a particle. Local maxima in the correlation image were detected with a custom-written peak finding routine. To achieve subpixel resolution, a 2-dimensional Gaussian function was fitted to every peak. Then, the particles were tracked with uTrack 2.3[100–102]. The derivative of the position vector was calculated to determine the instantaneous velocity of the particles, which was subsequently filtered with a symmetric median filter of $\pm 15\,\text{s}$ (which amounts to 31 subsequent frames) to reduce positioning noise. Particles were defined to be bound when the median filtered velocity was below $0.25\,\mu\text{m s}^{-1}$ and unbound when it exceeded this threshold. This threshold is well suited to separate both regimes as it minimizes the number of motion state changes both for highly adhesive and non-adhesive particles (see Supplementary Fig. 14). For further analysis, the trajectories were filtered with the following conditions to exclude tracking errors:

- Only particle trajectories which started during the sedimentation phase were used to exclude particles which happened to bind before the sedimentation phase.

- Only trajectories of particles which were unbound in the first frame were used. This excluded tracking errors in cases where a particle is lost and retracked, but already bound, which would lead to an underestimation of the fraction of irreversible binding events.

- Furthermore, only trajectories which did not end earlier than 5 s before the start of the rupture phase were used for evaluation. As by experiment design, particles never vanish during the sedimentation phase, this excludes further tracking errors when a particle was lost by the tracker during the sedimentation phase.

- To remove particle clusters, particles which were closer than $4.5\,\mu\text{m}$ to another particle for more than 30 s were excluded from the data analysis.

Examples of trajectories and the classification of their respective binding state are given in Supplementary Fig. 15.

To determine whether a particle is close to a cell or close to a coverslip, we used the convolutional neural net GoogLeNet[103] pre-trained with the ImageNet database[104]. We adapted the network to our needs using transfer learning[105] by replacing the 'loss3-classifier'-layer with a custom fully-connected-layer with an output size of 2, representing whether or not the particle is close to a cell. As input images, we cropped subimages of $52 \times 52\,\mu\text{m}^2$ around every particle, showing the particle in the center along with its nearest environment. The subimages were scaled to match the input size of GoogLeNet. In total, 1560 manually classified subimages of particles close to cells and 1560 manually classified subimages of particles close to the coverslip were

used for training (Supplementary Fig. 16) and about 400 images of each of the two categories were used as validation data. Both data sets were augmented by random reflection, rotation, scale, slight shear of up to 15°, and slight translation of up to $1.5\,\mu\text{m}$ to regularize the training process[106]. Adaptive moment estimation[107] was used to train the network. Particles manually classified to be close to a cell were classified identically in 96.6% of the cases by the network. Particles manually classified to be close to the coverslip were classified identically in 96.3% of the cases by the network. Most of the discrepancies were edge cases, which were also difficult to discriminate by eye.

To quantify the binding kinetics of the particles, the scheme illustrated in Fig. 2a was used. Whenever a particle switches from the unbound to the bound state, a binding event was registered. In the opposite case, an unbinding event was registered. We define the on rate $k_{\text{on,cell}}$, with which particles bind to the cells as the total number of binding events near a cell $N_{\text{binding,cell}}$ divided by the total time the particles are unbound $T_{\text{unbound,cell}}$ while near a cell:

$$k_{\text{on,cell}} = \frac{N_{\text{binding,cell}}}{T_{\text{unbound,cell}}}. \tag{24}$$

Analogously, the off rate $k_{\text{off,cell}}$ is defined by the number of unbinding events $N_{\text{unbinding,cell}}$ divided by the total time $T_{\text{bound,cell}}$, the particles are bound to a cell:

$$k_{\text{off,cell}} = \frac{N_{\text{unbinding,cell}}}{T_{\text{bound,cell}}}. \tag{25}$$

The on rate $k_{\text{on,coverslip}}$ and the off rate $k_{\text{off,coverslips}}$, which describe the binding kinetics between the particles and the coverslips are defined analogously. We observed that some particles never unbound. To quantify this behavior, binding events which started at least 200 s before and lasted until the end of the sedimentation phase were classified to be irreversible.

### Measuring the binding kinetics of microplastic particles

In each experiment, we used a low flow speed of $9\,\mu\text{L s}^{-1}$ to flush microplastic particles into microfluidic flow chambers and allowed the particles to sediment for 10 min without a flow. During this phase, we observed the binding and unbinding events between the particles and the cells as well as between the particles and the coverslips. Depending on the particle type, we observed different transition kinetics between the bound and the unbound state. We characterized the particles' binding kinetics by their respective binding and unbinding rates during the first 10 min after flushing (see Fig. 2c, d). $k_{\text{on}}$ and $k_{\text{off}}$ to and from cells and coverslips were determined and the results of the trajectories of all 9 independent experiments were averaged. Each experiment typically contained a few hundred independent trajectories.

To investigate whether components of the image medium adhered to the particles and thereby changed the adhesion between particles and cells or whether sticky particles got stuck in the tubing of the microfluidic device, we performed the same experiments 0, 40, and 80 min after the addition of the particles. We did not detect a time dependency of the results in most cases. Only in rare cases, a tendency to slightly weaker adhesion with time was found, which was reflected by a slightly increasing $k_{\text{on}}$ and a slightly decreasing $k_{\text{off}}$ with time (Supplementary Fig. 17). For this reason, we decided to pool the data of the experiments that were done 0, 40, and 80 min after the addition of the particles.

### Measuring the Adhesion strength of microplastic particles

To evaluate the adhesion strength between particles and cells as well as between particles and coverslips, we applied a constant hydrodynamic drag force of 50 pN to the particles after the sedimentation phase. To

remove any moving particles before the analysis of these experiments, a rolling median filter in time with a window size of 3 s was applied to the image sequence. The remaining static particles were then detected and classified as described above. We defined the fraction of particles that is still attached after 30 s of applied hydrodynamic force as the fraction of remaining particles.

Similar to the binding and unbinding rates, we found no significant differences in the relative attachment of particles to cells and to coverslips 0, 40, and 80 min after the addition of the particles, indicating that the incubation time is negligible in almost all cases (Supplementary Fig. 18). Therefore, we decided to pool for each particle type the data of the relative attachment of the three time points. After each measurement, the channels were disconnected from the microfluidic system and cleaned with trypsin, deionized water and 70% ethanol for 10 min each, removing cells and particles left in the channel. After each time series, the medium in the tubing of the microfluidic system was discarded, and the tubing was cleaned with deionized water and 70% ethanol to remove remaining particles in the system. The channels were reused three times with the same type of microplastic particles.

### Internalization experiment

The experiments were carried out as described in Ramsperger et al.[15]. In brief, prior to the experiments, the cells were scraped off the culture flask bottoms into the culture media, centrifuged ($200\times g$, 2 min, 20 °C) and re-suspended with 5 mL of cell culture medium in a Falcon tube (CORNING, Corning, New York, USA). Then, the cells were counted using a haemocytometer (Neubauer improved, Brand, Wertheim, Germany), seeded on microscope coverslips (diameter: 18 mm, #1, MENZEL GLAESER, Braunschweig, Germany) in 12-well plates (CellStar, GREINER BIO-ONE, Frickenhausen, Germany) in 1 mL of cell culture medium and allowed to adhere to the coverslips under standard culture conditions (37 °C, 5% $CO_2$, humidified) overnight. On each coverslip, 50,000 cells per mL were seeded to obtain a mean number of about 40,000 cells per coverslip (not all cells adhered to and remained on the coverslips during the preparation of the samples).

The following procedure was implemented to obtain samples for the quantification of microplastic particles interacting with cells (particle-cell-interaction), the measurement of the area covered by cells on the coverslips and for investigating the number of internalized microplastic particles from the particle-cell interactions. The 12-well plates containing the prepared cells were placed on ice for 1 h to reduce cellular activity. Due to different particle concentrations of the particle stock solutions, we diluted the stock solutions with PBS to obtain 150,000 beads per coverslip for each particle type. Three coverslips for each particle type were prepared, yielding a total of 33 coverslips. The particles were added to each coverslip and the experiment proceeded[15]. First, we quantified the number of particle-cell interactions and the area covered with cells on each coverslip within five regions of interest (ROIs) (0.29 mm²) using a DMI 6000 microscope (LEICA, Wetzlar, Germany, HCX PL APO 63× oil immersion objective, NA = 1.30) including a spinning disc unit (CSU-X1, YOKOGAWA, Musashino, Japan) with an EMCCD camera (Evolve 512, PHOTOMETRICS, Tucson, Arizona, including an additional 1.2× magnification lens). A differential interference contrast (DIC) microscopy image was acquired to quantify the particle-cell interactions within the ROIs using the Fiji ImageJ (version 1.53c) cell counter software. Additionally, spinning disk confocal stacks of fluorescently labelled cells were acquired using a 488 nm laser (50 mW, Sapphire 488, COHERENT, Santa Clara, California) at a spinning disc speed of 5000 rpm to excite fluorescence. Axial stacks of the cells were acquired with a vertical distance of 0.2 μm, which is sufficient to oversample the image given the axial resolution of the microscope[108].

To calculate the area covered by cells, both the DIC and fluorescence images were used[15]. First, a local contrast filter was applied to the

DIC images to approximate the cell mask $M_{DIC}$ at any given position $(i,j)$:

$$M_{DIC}(i,j) = \begin{cases} 1, \text{ if } \Delta I_{DIC} > T_{DIC} \\ 0, \text{ if } \Delta I_{DIC} \leq T_{DIC} \end{cases} \qquad (26)$$

The local intensity difference $\Delta I_{DIC}$ was evaluated within a circular region with a radius of 3 pixels. The threshold $T_{DIC}$ was chosen manually to optimize the cell detection. Next, the fluorescence images were evaluated to obtain the cell mask $M_F$. The maximum projection of each stack was calculated, and a manually chosen threshold $T_F$ was applied:

$$M_F(i,j) = \begin{cases} 1, \text{ if } I(i,j) > T_F \\ 0, \text{ if } I(i,j) \leq T_F \end{cases} \qquad (27)$$

To obtain the final cell masks, both individual masks were multiplied:

$$M(i,j) = M_{DIC}(i,j) M_F(i,j) \qquad (28)$$

Finally, small holes up to a size of 40 μm² were filled, and the masks were smoothed using a Gaussian filter with a radius of 3 pixels. Objects smaller than 80 μm² were excluded to reduce background noise. Finally, the area covered by cells within a ROI was extrapolated to the whole coverslip (245.5 μm²).

To quantify the number of particle-cell interactions, slight variations of the area covered by the cells and the number of microplastic particles was considered by standardizing each coverslip[15]. First, the measured number of particle-cell interactions were extrapolated to a whole coverslip, $PCI_{CS}$. Then, the number of cells on a standard coverslip $N_{cells,standardCS}$ was calculated by dividing the mean over all treatments of the area covered by cells on a coverslip by the area of an average single cell. The number of microplastic particles on a standard coverslip $N_{particles,standardCS}$ was the mean over all treatments of the number of particles added to the coverslips. Then, the number of particle-cell interactions on a coverslip was calculated:

$$PCI_{stand} = PCI_{CS} \left( \frac{N_{cells,standardCS}}{\frac{A_{CS}}{A_{singleCell,CS}}} \right) \left( \frac{N_{particles,standardCS}}{N_{particles,CS}} \right) \qquad (29)$$

With the measured number of particle-cell interactions on a single coverslip $PCI_{CS}$, the measured cell area on the coverslip $A_{CS}$, the average area of a single cell on the coverslip $A_{singleCell,CS}$, and the number of particles added to the coverslip $N_{particles,CS}$.

After quantification of the particle-cell interactions, we measured the conditional internalization probability. From the same samples used to quantify particle-cell interactions and areas covered with cells on coverslips, we visually screened each sample for single particle-cell interactions to distinguish between particles that were only attached to cells and particles that were internalized by cells. The above mentioned DMI 6000 microscope with a higher magnification (100× oil immersion objective, NA = 1.40) was used here. Beginning from a randomly defined starting point, the coverslips were screened in the DIC-channel until 100 particle-cell interactions were detected. Once a particle-cell-interaction was found, a DIC-image was taken, and axial confocal stacks of fluorescently labelled cells were acquired (vertical distance of the axial stacks: 0.2 μm). To evaluate internalization of the microplastic particles, each confocal stack of cells with fluorescently labelled actin filaments was analyzed with Fiji ImageJ (version: 1.53c) orthogonal views. The microplastic particles used in the experiments were non-fluorescent and therefore not directly visible in the confocal stacks. The DIC images were used to mark the particle positions (using the ROI manager in Fiji ImageJ). These positions were then transferred

to the confocal stacks, in which internalized particles were visible as spherical black regions within the actin network. Only microplastic particles that were fully surrounded by actin filaments were considered to be internalized. Microplastic particles that were only partly surrounded were considered to be attached to the cells. Finally, the internalization probability was calculated as the ratio of internalized particles to the number of particle-cell interactions. For the calculation of the internalization probability for particles coated with an ecocorona, namely salt and freshwater particles incubated for two and four weeks the data published by Ramsperger et al.[15] were used.

## Internalization mechanisms

To test our hypothesis that the microplastic particles were internalized via an actin-dependent mechanism such as phagocytosis or macropinocytosis, we performed experiments with live cells, monitoring the actin dynamics during particle internalization and the subsequent acidification of the particles. Two days before an experiment, $5 \times 10^4$ J774A.1 cells stably transfected with a LifeAct-GFP construct were seeded on 18 mm glass coverslips (MENZEL GLAESER, Braunschweig, Germany) in a 12-well plate (CellStar, GREINER BIO-ONE, Frickenhausen, Germany) containing 1 mL of cell culture medium. 30 min before an experiment, the medium was exchanged with cell culture medium containing $0.1\,\mu L\,mL^{-1}$ (final concentration of 100 nM) of LysoTracker Red DND99 (Thermo Fisher Scientific Inc, Waltham, MA). Right before the experiment, the coverslips were mounted on a custom aluminum sample holder. The cells were covered with $84\,\mu L$ of imaging medium containing $3\,\mu L\,mL^{-1}$ of the corresponding bead stock solution, and the sample was covered with another 18 mm coverslip. The sample was immediately mounted on the fluorescence microscope and live cell imaging started.

Imaging was performed with a frame rate of 0.32 Hz on an inverted, motorized microscope (Nikon Eclipse Ti-E, Nikon, Tokyo, Japan) with a 40× water immersion objective (CFI Apo LWD 40× WI λS, Nikon, NA = 1.15), which was equipped with a EMCCD camera (Andor Luca R, Oxford Instruments, Belfast, United Kingdom). The microscope body was enclosed in a custom-built incubation chamber which keeps the body of the microscope and the sample at a temperature of 37 °C. The imaging modes were either brightfield illumination or widefield epi-fluorescence microscopy (Nikon Intensilight). For each frame, we imaged 3 channels: Brightfield (exposure time 100 ms), GFP-L (LifeAct channel, exc. 460–500 nm, em. >510 nm; exposure time 700 ms), and Texas Red (LysoTracker channel, exc. 540–580 nm, em. 600–660 nm; exposure time 500 ms). Imaging was continued for 45 min, leading to a total of 864 acquired images.

For the evaluation of individual internalization and maturation events, particles were tracked in the brightfield channel using a custom tracking algorithm based on radial symmetry[109]. Only particles that stayed in the focal plane of the objective were considered for evaluation. LifeAct and LysoTracker intensities were evaluated using MATLAB. Based on the particle trajectories from the brightfield channel, the images in the LifeAct and LysoTracker channels were cropped to a ROI of 60×60 pixels around the particle. The radius of each individual microplastic particle was determined, to account for the dispersity in size of some particle types. The particle radius $r$ in pixels (px) was chosen in a way that maximized the signal-to-background ratio of both fluorescence channels. Then, the average LifeAct and LysoTracker intensities were evaluated for each frame in a ring-shaped ROI in the interval $[r - 3\,px, r + 2\,px]$, around the particle surface. This intensity was normalized to the average intensity of a ring-shaped ROI further apart from the particles surface, in the interval $[r + 9\,px, r + 12\,px]$. This normalization of the LifeAct and LysoTracker intensities corrected for artifacts like photobleaching. Furthermore, it enabled the evaluation of very localized actin dynamics and acidification around the microplastic particles' surface since global changes of fluorescence intensity did not contribute.

## Statistical analysis

Statistical analysis was conducted using R studio software (version 4.0.2, 2020-06-22) with the packages: "car", "carData", "rstatix", "multcompView". To test for significant differences between the particle types, and whether there was a significant time dependence, the data for the relative attachment were tested for normal distribution (Shapiro–Wilk test) and homogeneity of variances (Levene test). If the Shapiro–Wilk test or the Levene test were significant, a two-sided Kruskal–Wallis test with a Games Howell post hoc test was conducted to check for differences between microplastic particle types. Otherwise, a one-way ANOVA with a Tukey post hoc test was performed. Correlation coefficients (Pearson's $R$) were calculated using MATLAB. A detailed summary of all statistical tests is provided in Supplementary Data 1.

## Reporting summary

Further information on research design is available in the Nature Portfolio Reporting Summary linked to this article.

## Data availability

The data that support the findings of this study are available via Zenodo[110] and from the corresponding authors upon request. Source data are provided with this paper.

## Code availability

All code used in the analysis is available via Zenodo[110] and from the corresponding authors upon request.

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

## Acknowledgements

We thank Thomas Scheibel and Hendrik Bargel for their support with the SEM. We thank Alexander Rohrbach and Rebecca Michiels for providing the LifeAct-GFP-transfected J774A.1 murine macrophages. This work was supported by the Deutsche Forschungsgemeinschaft (DFG, German Research Foundation) – project number 391977956 – SFB 1357 and received funding from the European Union's Horizon2020 Research and Innovation programme, under the Grant Agreement number 965367 (PlasticsFatE). The SEM was funded by the Deutsche Forschungsgemeinschaft (DFG GZ: INST 91/366–1 FUGG and INST 91/427–1 FUGG). AFRMR was supported by a scholarship of the elite network of Bavaria (BayEFG). SW and ML were supported by the elite network of Bavaria (Study Program Biological Physics). WG, AFRMR, and SW were supported by the University of Bayreuth Graduate School. MO was supported by the Deutsche Forschungsgemeinschaft (DFG, OB362/4-1). Part of the research described in this paper was performed at the Canadian Light Source, a national research facility of the University of Saskatchewan, which is supported by the Canada Foundation for Innovation (CFI), the Natural Sciences and Engineering Research Council (NSERC), the National Research Council (NRC), the Canadian Institutes of Health Research (CIHR), the Government of Saskatchewan, and the University of Saskatchewan.

## Author contributions

H.K., C.L., W.G., A.F.R.M.R. and S.W. initiated the research. H.K., C.L., A.F., G.K.A., W.G., A.F.R.M.R. and S.W. planned the research. W.G., M.L., S.G. and H.K. developed the microfluidic method. S.W., A.F.R.M.R., W.G., T.W., A.C., and M.O. performed the experiments. W.G., A.F.R.M.R., and S.W. wrote the draft of the manuscript. All authors revised and edited the manuscript.

## Funding

## Competing interests

The authors declare no competing interests.

## Additional information

**Supplementary information** The online version contains Supplementary Material available at https://doi.org/10.1038/s41467-024-45281-4.

