## [Peer Review File · Nature Communications]

Nominally identical microplastic models differ greatly in their particle-cell interactionsReviewers' comments:

Reviewer #1 (Remarks to the Author):

The authors submitted a manuscript entitled " Same but different: the choice of model microplastics strongly affects particle-interactions" in which they investigated a range of commercially available spherical particles made of polystyrene with an average diameter of 3 μm for cell adhesion and internalization probability, while also investigating the effects of incubation in different waters leading to the formation of an eco-corona and its influence on these parameters.

Their main conclusion is that cell adhesion and internalization of the particles correlate with their zeta potential. On the one hand, their study is very commendable because they comprehensively show that a detailed analysis of the particles is a basic prerequisite for data interpretation and that, unfortunately, in a myriad of studies, PS particles are used in a relatively unreflective and insufficiently characterized manner. On the other hand, the main statement that the interaction of particles with cells depends on their zeta potential is not a new finding and has been known in the particle research field for a very long time.

In addition, there are a few blurbs in the manuscript that should be corrected. For example, an increase in reactive oxygen species does not equal a cytotoxic effect or a particle-cell interaction does not equal significant cellular effects. Especially with materials that the cells are not familiar with, they often do not have the necessary mechanisms to process them, such as the degradation of plastic or starch. For these reasons, the zeta potential, as shown by the authors, is also an important parameter for microplastic particles in order to assess possible cell/particle interactions, but is by no means sufficient to carry out a credible risk assessment, since each cell type, each single cell has a different surface, receptor equipment, etc. and thus a separate risk assessment would have to be carried out for each cell type, in addition, some other parameters are essential for the interactions (shape of the particles, most microplastics found in the environment are not spherical, material, possibly the same zeta potential, etc.).

It is also not completely clear, why the authors used the absolute values of the zetapotential to show the correlations. Especially in figure 3b it become apparent that the positively charged amino-particles have also the highest absolute internalization probability which fits with the proposed interaction of these particles with the negatively charged cell surface and contradicts a bit the statement that negatively as well as positively charged particles might interact/adhere in a similar manner. The work with the water-treated particles will be hardly reproducible just because of the accessibility of the used water sources and the not comprehensively analysed/described water in their previous publication (Sci. Adv.). I cannot agree with the statement, that internalization generally needs a receptor (line 296), because it is known that electrostatic forces can lead to adsorption mediated endocytosis. Maybe this should be described clearer and I have misunderstood the statement.

In conclusion, the publication is important for the microplastics field because it very nicely shows that a detailed characterization of the used material is essential for understanding and interpretation of the data. Also the established microscopic pipeline is a very nice tool, but the major outcome is not really novel and expectable.

Reviewer #2 (Remarks to the Author):

I think it is really good to be open minded and allow space for new expressions of old ideas to find a new way. That advances science. But there must be some reasonable reach back to existing knowledge or we lose any reasonable concept of scholarship.

In those respects I would very much like to be supportive, but am struggling with the issues a bit. The field of particulate and Nanosciences measured and analysed the surfaces of plastic particles ad infinitum, to the degree that they were synthesised to control this. The fact that manufacturers particles were not standardised, and in fact varied from batch to batch was an long discussion, and much was understood. The need and role of environmental adsorbents (coronas) was also discussed at length, and also the nature of the interactions of bare or nearly bare surfaces on organisms (*Journal of the American Chemical Society*, 135 (4): 1438-1444, 2013)). Much debate showed how this could lead to many anomalies., *Nanomedicine: Nanotechnology, Biology and Medicine* Volume 9, Issue 8, November 2013, Pages 1159-1168). There were dozens of such papers, and the above is a random sample, even of those I know.

If one starts to cite too many articles then one risks being unfair to some, but I think the basic point is clear that one has to think carefully about making the same points over again, in some cases with much less detail and data, in a new arena.

This has to be balanced with the need for new thrusts into the fields to invite new questions, and the need for new people to have space to grow, get funded, and learn. I would not lightly reject this paper. Possibly the useful things to do is invite the authors to consider all this themselves. Certainly these are respectable and thoughtful authors and I feel they might be able to pilot a course for their own endeavours, and explain it to others.

Reviewer #3 (Remarks to the Author):

In the current study, Gross et al, have developed a platform to correlate the microparticle surface charge to particle- cell adhesion strength and possibly predict a biological response. The correlation between particles and the physico-chemical properties and their behavior in environmental and biological media is still under debate and needs to be addressed, therefore this article is of interest and timely.

In this study, the authors have chosen a range of commercially available microparticles of 3 microns in nominal size but with different surface properties (plain, carboxylated and aminated) and have also evaluated the change of zeta potential after exposure salty and fresh water.

The authors also highlighted another important factor, which is that all microparticles of the same types are not the same, and they may behave differently and they have included in the study plain microparticles from eight different sources, two carboxylated and one amino modified.

in the study, they have used a microfluidic system which offers an interesting tool to measure the internalization probability and binding affinity which measured the K_{on} / K_{off} and binding strength.

the reviewer feels that the study is elegant and systematic, and the preliminary data with the environmental media has the potential to provide useful info to environmental toxicology. However, the reviewer has identified the following gaps:

1. Batch to batch variation and microparticles of the same type are likely to be significantly different. the authors decided to focus on the zeta potential changes only, but the surface morphology can also play a role in the colloidal stability and eco-corona formation. This aspect should be taken into consideration.
2. the zeta potential is significantly different also across the same microparticle types, which is likely to be caused by the use of a capping agent that can increase the particle stability. this might be more relevant for plain microparticles which are likely to have a neutral surface charge and poor colloidal stability. the presence of a capping agent can also be affected when the particles are diluted in the cell culture media prior to the microfluidic exposure. the authors should consider picking the material from a unique source and applying a chemical modification to have identical material core and different surface properties.
3. the colloidal stability has to be measured and provide evidence that no rapid sedimentation occurs. The experimental settings require the particles to be diluted in the media used to evaluate the cell binding. SEM analysis was also measured on dry samples and gives no information on the colloidal stability.

4. it is not clear the choice of the cell type and the link with the eco-tox
5. The experimental part is detailed enough. However, the reviewer could not easily find the procedure for the preparation of the working solution of the microparticles for the binding studies, the composition of the media used for the eco-corona (salt water, fresh water) and the sample prep. While the experimental settings are referenced, the main experimental features should be mentioned again. The reviewer also struggles to understand the relevance of the eco-corona when using murine cell lines.
6. the MM zeta potential varied when using different media, the reviewer suggests to further characterising the eco-corona so that it could help understand the zeta-potential change of the eco-corona. the authors refer to a previous publication but, this info should be provided also in this article.
7. the internalization experiment needs to be validated with another technique (eg flow cytometry or SPR or QCM) to increase its robustness
8. The reviewer struggles to appreciate figure 1 as it is too packed and the main results are lost. is figure 1A a cross section of figure 1B? how was it made? the figure is packed and there are too many “take home messages” which is the variance of the binding affinity when changing the microparticle surface property, surface charge and the presence of eco-corona. the reviewer suggests simplifying it, and moving some data in the SI (eg coverslip binding or similar behaviour of microparticles with same surface charge or same nominal properties).

Reviewer #4 (Remarks to the Author):

There are some nice techniques and methods presented here - I especially like the microfluidic methods to determine the on/off rates and the irreversible binding parameters and the particle uptake analysis.

I do however have a number of major concerns regarding various elements of the work, its overall meaning and the understanding and conclusions being drawn.

(i). Zeta potential is known to change dramatically in serum with hard and soft corona's attached to the particles. The Zeta potential of the particles used here needs to be determined in serum and/or media in which the cells are grown/analysed. I do not see much relevance in equating the Zeta potentials of particles in KCl to how they behave with cells in media (see for example the paper “Surface functionalization affects the zeta potential, coronal stability and membranolytic activity of polymeric nanoparticles in which a series of particles of different Zeta potentials/charges all had the same Zeta potential when in biological solutions.

(ii). The corona on particles is typically based on irreversible binding of proteins to give a hard corona and then a more dynamic “soft” corona. How do the particles form a so-called eco-corona? Do the particles “swell” or uptake the buffer/ionic solution? How long would this take ? Would it wash out when the Zeta potentials are measured (this is done in KCl) or when the solvent/buffer is changed? As such I am not convinced by the meaning/measurements of the Zeta potentials of the particles stored in freshwater or salt water and then measured in KCl. I think changes in time of the measurements after removal from freshwater or salt water (i.e. how long they are in the KCl Zeta potential measurement solutions) should be explored.

(iii). If the particles are left in fresh water or salt water for days is there algae growth that generates proteins that bind to the particles? This is quite common in bio-fouling.

(iv). The beads will be stabilised by surfactants etc.... as the authors say. How stable are these? Are the changes in Zeta potential (e.g. following storage in fresh water) due to the loss of the stabiliser rather than anything else? This is key as the unmodified beads (for example ST) have a Zeta potential that is as -ve as the carboxylate beads which is surprising. In addition, even the unmodified particles might well be stabilised with neutral surfactants such as PVA etc... Is it this that binds to cells or proteins?

(v). As a general rule +vely charged particles/molecules interact with cells much more readily than -vely charged ones (due to the nature of the cell membrane) - This is opposite to that seen here. Please explain. The whole section on their claims that local electrostatic interactions and binding with the cell membrane is flawed - the Zeta potential (in media) has not been determined but I would imagine they will be pretty similar across all the particles.

(vi). The fact that cell binding and coverslip binding show the same trends and similar on/off rates for the particles worries me. What is the nature of interaction on the coverslips? Is the particle just getting “stuck” in defects on the cover slip surface? i.e. they say “particles adhered strongly to cells and coverslips...”. What is the nature of the coverslip, is it getting coated with protein too ? Is this what is controlling particle binding ?

(vii). Figure 3. The sign of the Zeta potential on the particles (e.g. +29 mV for the amine (+vely charged particles and -91 mV for the negatively acid particles) is ignored in the graphs. I do not think it is OK to put +ve and -ve Zeta values on a log plot and hence ignore the sign. Cells respond so differently to different charged particles. Of course, they could all have the same Zeta potential (due to a protein corona) but then we would expect to see few differences between the particles.

(viii). Figure 3 is a log/log plot. As such I am not convinced that there is any relationship here and I am not sure why a log scale is being used or relevant.

Minor general points

Label the particles 1-11.... Having a list of abbreviations helps little and makes it very cumbersome.

The density of the particles needs to be measured as this might well also affect binding.

Use of the word "plain" for particles - "unmodified" would be better.

Please don't repeat and over-labor trivial points e.g. kon/koff etc...

Particle clumping behavior should be mentioned - often an issue with low Zeta potential particles.

SI: Images for MM-SW2/4 show salt crystals. I would suggest this is due to lack of washing (as the beads have been soaked in salt water) rather than a real feature of the beads.

Letter of Response

Reviewer #1

The authors submitted a manuscript entitled " Same but different: the choice of model microplastics strongly affects particle-interactions" in which they investigated a range of commercially available spherical particles made of polystyrene with an average diameter of 3 μm for cell adhesion and internalization probability, while also investigating the effects of incubation in different waters leading to the formation of an eco-corona and its influence on these parameters.

Their main conclusion is that cell adhesion and internalization of the particles correlate with their zeta potential. On the one hand, their study is very commendable because they comprehensively show that a detailed analysis of the particles is a basic prerequisite for data interpretation and that, unfortunately, in a myriad of studies, PS particles are used in a relatively unreflective and insufficiently characterized manner. On the other hand, the main statement that the interaction of particles with cells depends on their zeta potential is not a new finding and has been known in the particle research field for a very long time.

We thank Reviewer #1 for the helpful comments on our manuscript. In our revised manuscript, we carefully considered these comments to clarify our results and resolve the reviewer's concerns. We are aware that a lot of research about the zeta potential and cellular interactions has been done in the field of particle toxicology. For example, in the nanoparticle field it is well established that cellular interactions and internalization depend on the particles' zeta potential. However, these results from *nanoparticle* research cannot be directly transferred to *microparticles*, since their surface-to-volume ratio differs tremendously, and since nano- and microparticles are internalized into cells by completely different mechanisms. Furthermore, research about the role of the zeta potential for cellular interactions and internalization of microparticles is less conclusive. For example, some studies previously showed that less negatively charged microparticles interact with cells more often, whereas other studies showed that both negatively and positively charged microparticles were internalized efficiently, and that an increase in negative surface charge can lead to an increase in internalization efficiency. Also, these studies often only investigated a few different microparticles in a narrow range of zeta potentials.

In our study, we therefore systematically analyzed the cellular interactions of twelve different microparticle types spanning a wide range of zeta potentials from -4.7 mV to -93.1 mV. With our novel microfluidic approach, we individually assessed the role of the zeta potential for the microparticle binding kinetics, adhesion strength, conditional internalization probability, and absolute internalization probability. In this way, we were able to assess the role of the zeta potential for cellular interactions and internalization in a quantitative way that was not achieved in previous studies. Furthermore, we also investigated the internalization mechanism for these particles. Therefore, we think that our study is not only relevant to the microplastics community, but also for the field of particle toxicology as well as for biophysics and cell biology.

To highlight the novelty of our study, we changed the introduction accordingly. We reviewed the existing literature on the role of the zeta potential for the cellular interactions of nano- and microparticles more thoroughly (lines 75-105) and highlighted the existing knowledge gaps (lines 97 - 112). Furthermore, in the discussion part of the manuscript, we emphasized the originality of our results and our microscopic pipeline (lines 331-339 and 467-473).

In addition, there are a few blurbs in the manuscript that should be corrected. For example, an increase in reactive oxygen species does not equal a cytotoxic effect or a particle-cell interaction does not equal significant cellular effects. Especially with materials that the cells are not familiar with, they often do

not have the necessary mechanisms to process them, such as the degradation of plastic or starch. For these reasons, the zeta potential, as shown by the authors, is also an important parameter for microplastic particles in order to assess possible cell/particle interactions, but is by no means sufficient to carry out a credible risk assessment, since each cell type, each single cell has a different surface, receptor equipment, etc. and thus a separate risk assessment would have to be carried out for each cell type, in addition, some other parameters are essential for the interactions (shape of the particles, most microplastics found in the environment are not spherical, material, possibly the same zeta potential, etc.).

We thank the reviewer for pointing out these inaccuracies in our manuscript. We agree that the zeta potential alone is by no means sufficient to assess the cytotoxicity or evaluate the risk associated with a microplastic particle. For risk assessment purposes, the complex interactions of physical and chemical particle properties, as well as biological fate of internalized particles have to be considered.

However, we show in our study that the zeta potential determines the binding affinity, adhesion strength, and absolute internalization probability of microplastic particles. The interaction and internalization of a microplastic particle are by no means a cellular effect. However, they are one prerequisite for subsequent processes leading to cellular effects and cytotoxicity. Considering this, the zeta potential of microplastic particles is one property that has to be taken into account in risk assessment studies. To clarify this in our manuscript, we updated the respective parts (lines 64-65, 446-450, 470-473).

It is also not completely clear, why the authors used the absolute values of the zetapotential to show the correlations. Especially in figure 3b it become apparent that the positively charged amino-particles have also the highest absolute internalization probability which fits with the proposed interaction of these particles with the negatively charged cell surface and contradicts a bit the statement that negatively as well as positively charged particles might interact/adhere in a similar manner.

We appreciate this observation of the reviewer. In the previous version of the manuscript, we included 3 different functionalized particle types, one of which was amine-modified. These functionalized particles behaved very similar to the unmodified particles and followed the correlations of the binding kinetics, adhesion strength, and absolute internalization probability very well. However, we now decided to leave the functionalized particles out of this study for several reasons.

First, we agree with the reviewer that drawing statistically significant conclusions regarding the influence of charge polarity on cellular interactions and internalization is challenging, since we only included one positively charged particle type. Second, introducing a wider array of functionalized particle types would introduce additional complexity in interpreting the results. Particle-cell interactions are likely influenced by other factors beyond just charge, such as the specific chemical identity of the functional groups that may facilitate specific receptor-ligand interactions.

We agree with the reviewer that it might be interesting to assess the effect of different functional surface groups and charge polarities of microplastic particles for their interactions with environmental and biological media, and their cellular interactions and internalization. This may become particularly relevant in the context of ecotoxicology, where environmental plastic particles often carry functional surface groups, e.g., acquired through aging and weathering processes. However, we believe that this is beyond the scope of this study, which primarily focuses on the role of the zeta potential for particle-cell interactions of nominally identical, unmodified model microplastic particles.

The work with the water-treated particles will be hardly reproducible just because of the accessibility of the used water sources and the not comprehensively analysed/described water in their previous publication (Sci. Adv.).

The environmental media used in our study were taken from an outside freshwater pond close to the University of Bayreuth and a large seawater tank simulating a coral reef ecosystem. Both ecosystems are likely very specific in their composition of species, nutrients, and mineral contents. Especially the outside freshwater pond is additionally subjected to seasonal variations, making the contents of the environmental media used in our study season dependent. Thus, we agree with the reviewer that these environmental media are unique to our study.

However, our results indicate that exposure to environmental media from two ecosystems that could not be more different (middle European freshwater pond vs. tropical coral reef) lead to the formation of an eco-corona on the particles that affects their zeta potential and consequently particle-cell interactions in a very similar way (lines 186-189, 221-237, 256-264, and 294-298). To better characterize the environmental media and their effect on the microplastic particles, we additionally tried to identify key species of the media (Supplementary Figure 4), and further analyzed the resulting eco-corona quantitatively using scanning electron microscopy (lines 157-164, Supplementary Figure 1, Supplementary Table 1), synchrotron-based scanning transmission X-ray microscopy (lines 168-176, Supplementary Figure 5, Supplementary Table 2), and X-ray photoelectron spectroscopy (lines 177-86, Supplementary Table 3).

I cannot agree with the statement, that internalization generally needs a receptor (line 296), because it is known that electrostatic forces can lead to adsorption mediated endocytosis. Maybe this should be described clearer and I have misunderstood the statement.

We agree with the reviewer that internalization in general does not need a receptor and thank the reviewer for pointing out this misunderstanding. We additionally performed experiments with living cells transfected with a LifeAct-GFP construct to monitor actin dynamics during internalization. The cells were stained with LysoTracker dye to evaluate the maturation and acidification of particles after internalization (lines 303-323, Supplementary Figures 8 & 9). These experiments showed that all particles that were acidified inside the cell were internalized via an actin-dependent pathway (Supplementary Figure 10). This indicates that microplastic particles were internalized via phagocytosis or macropinocytosis.

While phagocytosis is tightly controlled by receptor-ligand interactions, macropinocytosis is a stochastic process that does not necessarily require activation of receptors. However, some receptors like EGFR lead to an increase in membrane ruffling upon activation, which increases the probability of macropinocytosis. In that sense, also macropinocytosis is not completely independent of receptor-ligand interactions. We clarified the respective parts in our manuscript (lines 432-437).

In conclusion, the publication is important for the microplastics field because it very nicely shows that a detailed characterization of the used material is essential for understanding and interpretation of the data. Also the established microscopic pipeline is a very nice tool, but the major outcome is not really novel and expectable.

We thank the reviewer again for the helpful and constructive comments on our manuscript. We understand the concerns about the novelty of our findings, as there are numerous publications about the role of the ζ -potential for nanoparticle-cell interactions, and several publications about the role of the ζ -potential for microparticle-cell interactions. However, as mentioned before, results from nanoparticles cannot simply be transferred to microparticles and microplastics, because they interact differently with cells due to their different surface-to-volume ratio, and the mechanisms of internalization strongly differ between nano- and microparticles. Furthermore, although previous research on microparticles shows that there is a relation between the ζ -potential and microparticle-cell interactions, the exact role of charge polarity and magnitude of the ζ -potential for particle-cell interactions and internalization is not unanimously clear yet. For example, some studies previously

showed that less negatively charged microparticles interact with cells more often, whereas other studies showed that both negatively and positively charged microparticles were internalized efficiently, and that an increase in negative surface charge can lead to an increase in internalization efficiency. Also, these studies often only investigated a few different microparticles in a narrow range of zeta potentials.

Our study provides valuable insight by systematically investigating twelve different particle types with a ζ -potential range spanning two orders of magnitude. With our approach, we were able to individually assess the ζ -potential's role for the particle binding kinetics, adhesion strength, conditional internalization probability, and absolute internalization probability. Therefore, our results enable a quantitative understanding of the ζ -potential's role for particle adhesion and internalization that has not been achieved by any previous study.

Reviewer #2

I think it is really good to be open minded and allow space for new expressions of old ideas to find a new way. That advances science. But there must be some reasonable reach back to existing knowledge or we lose any reasonable concept of scholarship.

*In those respects I would very much like to be supportive, but am struggling with the issues a bit. The field of particulate and Nanosciences measured and analysed the surfaces of plastic particles ad infinitum, to the degree that they were synthesised to control this. The fact that manufacturers particles were not standardised, and in fact varied from batch to batch was an long discussion, and much was understood. The need and role of environmental adsorbents (coronas) was also discussed at length, and also the nature of the interactions of bare or nearly bare surfaces on organisms (*Journal of the American Chemical Society*, 135 (4): 1438-1444, 2013)). Much debate showed how this could lead to many anomalies., *Nanomedicine: Nanotechnology, Biology and Medicine* Volume 9, Issue 8, November 2013, Pages 1159-1168). There were dozens of such papers, and the above is a random sample, even of those I know.*

If one starts to cite too many articles then one risks being unfair to some, but I think the basic point is clear that one has to think carefully about making the same points over again, in some cases with much less detail and data, in a new arena.

This has to be balanced with the need for new thrusts into the fields to invite new questions, and the need for new people to have space to grow, get funded, and learn. I would not lightly reject this paper. Possibly the useful things to do is invite the authors to consider all this themselves. Certainly these are respectable and thoughtful authors and I feel they might be able to pilot a course for their own endeavours, and explain it to others.

We thank Reviewer #2 for the insightful perspective on our manuscript. Sometimes it can be helpful to take a step back and look at the own work in a broader context. In this way, one can identify how one's own work can help to advance the field in a meaningful way.

We are grateful for the recommended literature about nanoparticle-cell interactions and were happy to include the references in our manuscript (lines 79, 309, 425). We are aware that a lot of research about cellular interactions of nanoparticles has been done in the past. For example, it is well established that cellular interactions and internalization depend on the nanoparticles' zeta potential and surface charge. We reviewed the existing literature more thoroughly in our introduction (lines 75-85). However, although these results from nanoparticle research are helpful to inform further research about microparticles and microplastics, they cannot be easily transferred to microparticles and

microplastics because of their different surface-to-volume ratio and completely different mechanisms of cellular internalization (lines 97-105).

There were also already some efforts to investigate the role of the zeta potential for microparticle-cell interactions, however, the results were less conclusive compared to nanoparticle research. For example, some studies previously showed that less negatively charged microparticles interact with cells more often, whereas other studies showed that both negatively and positively charged microparticles were internalized efficiently, and that an increase in negative surface charge can lead to an increase in internalization efficiency. Also, these studies often only investigated a few different microparticles in a narrow range of zeta potentials. We highlighted the existing research and consequential knowledge gaps in the introduction (lines 86-96).

Especially in the microplastics community there is a low awareness for the potential role of the zeta potential for microparticle-cell interactions. For example, of 216 studies about effects of microplastic particles on aquatic and mammalian models currently listed in the ToMEx database, only 17 % provided the zeta potential of the microplastic particles that were used. In the upcoming updated version of the database (not yet publicly available), even less (15%) of the included studies provided the zeta potential of the microplastic particles. We discussed this knowledge gap in the introduction (lines 106-112)

In our study, we therefore systematically analyzed the cellular interactions of twelve different microparticle types spanning a wide range of zeta potentials from -4.7 mV to -93.1 mV. With our novel microfluidic approach, we individually assessed the role of the zeta potential for the microparticle binding kinetics, adhesion strength, conditional internalization probability, and absolute internalization probability. In this way, we were able to assess the role of the zeta potential for cellular interactions and internalization in a quantitative way that was not achieved in previous studies. Furthermore, we also investigated the internalization mechanism for these particles. Therefore, we think that our study is not only relevant to the microplastics community, but also for the field of particle toxicology as well as for biophysics and cell biology. We emphasized the originality of our results and our microfluidic platform in the discussion part (lines 335-343 and 443-446).

Furthermore, to better understand the role of adsorbents (coronas) for microparticle-cell interactions and internalization, we further characterized the microplastic particles exposed to environmental media (lines 153-186, Supplementary Figures 1, 4, and 5, Supplementary Tables 1, 2, and 3). To assess how the particle properties change upon incubation in cell culture media and the potential role of protein coronas, we measured their zeta-potential before and after incubation in cell culture media (lines 145-152, 188-191, Supplementary Table 1, Supplementary Figure 2).

We also thank the reviewer for mentioning the importance of batch-to-batch variations when assessing the interactions of model microplastics or microparticles with cells. Although this was not the main point of our study, we highlighted the potential for batch-to-batch variations in the discussion part and emphasized the importance of a thorough characterization of model microplastic particles (lines 330-334, 457-459).

Reviewer #3

In the current study, Gross et al, have developed a platform to correlate the microparticle surface charge to particle- cell adhesion strength and possibly predict a biological response. The correlation between particles and the physico-chemical properties and their behavior in environmental and biological media is still under debate and needs to be addressed, therefore this article is of interest and timely.

In this study, the authors have chosen a range of commercially available microparticles of 3 microns in nominal size but with different surface properties (plain, carboxylated and aminated) and have also evaluated the change of zeta potential after exposure salty and fresh water.

The authors also highlighted another important factor, which is that all microparticles of the same types are not the same, and they may behave differently and they have included in the study plain microparticles from eight different sources, two carboxylated and one amino modified.

in the study, they have used a microfluidic system which offers an interesting tool to measure the internalization probability and binding affinity which measured the K_{on} / K_{off} and binding strength.

the reviewer feels that the study is elegant and systematic, and the preliminary data with the environmental media has the potential to provide useful info to environmental toxicology. However, the reviewer has identified the following gaps:

1. Batch to batch variation and microparticles of the same type are likely to be significantly different. the authors decided to focus on the zeta potential changes only, but the surface morphology can also play a role in the colloidal stability and eco-corona formation. This aspect should be taken into consideration.

We thank Reviewer #3 for the constructive feedback on our manuscript. We agree about the importance of batch-to-batch variations when assessing the interactions of model microplastics or microparticles with cells. Although this was not part of our study, we highlighted the potential for batch-to-batch variations in the discussion part and emphasized the importance of a thorough characterization of model microplastic particles (lines 330-334, 457-459).

2. the zeta potential is significantly different also across the same microparticle types, which is likely to be caused by the use of a capping agent that can increase the particle stability. this might be more relevant for plain microparticles which are likely to have a neutral surface charge and poor colloidal stability. the presence of a capping agent can also be affected when the particles are diluted in the cell culture media prior to the microfluidic exposure. the authors should consider picking the material from a unique source and applying a chemical modification to have identical material core and different surface properties.

We agree with the reviewer that surfactants may play a significant role for the zeta potential of the microplastic particles. Incubation in environmental media and cell culture might thus change the zeta potential of the microplastic particles by “washing out” the surfactant. To account for this effect in our experiment, we measured the zeta potential before and after 2h incubation of the particles in cell culture media (lines 145-152 and 188-191, Supplementary Table 1, Supplementary Figure 2).

Our results indicate that the magnitude of the zeta potential decreases after an incubation in cell culture media, possibly due to the formation of a protein corona or potential loss of surfactant. However, the differences in the zeta potential were preserved, so that particles with an initial zeta potential close to zero were still almost neutral after incubation, and negatively charged particles were still negative after incubation (Supplementary Figure 2). Overall, the zeta potential before and after incubation in cell culture media was strongly correlated.

We appreciate the suggestion of the reviewer to include more particles with a defined chemical surface modification. In the previous version of the manuscript, we included 3 different functionalized particles, two of which were carboxylated, and one of which was amine-modified. These functionalized particles behaved very similar to the unmodified particles. Functionalized particles that were strongly charged interacted strongly with the cells and were more frequently internalized, similar to the unmodified particles. However, we decided to leave them out of this study for several reasons.

First, only one of these functionalized particles had a positive zeta potential. Therefore, it is very difficult to draw statistically significant conclusions about the role of charge polarity for cellular interactions and internalization of the microparticles.

Apart from that, the inclusion of more functionalized particle types would introduce additional complexity in interpreting the results. Particle-cell interactions are likely influenced by other factors beyond just charge, for example the chemical identity of the functional groups that could lead to more specific receptor-ligand interactions.

We agree with the reviewer that it might be interesting to assess the effect of different functional surface groups and charge polarities of microplastic particles for their interactions with environmental and biological media, and their cellular interactions and internalization. This might also be relevant in an ecotoxicological context where environmental plastic particles carry functional surface groups, e.g., due to environmental weathering. However, we think that this falls outside of the scope of this study, which primarily focuses on the role of the zeta potential for particle-cell interactions of nominally identical, unmodified model microplastic particles.

3. the colloidal stability has to be measured and provide evidence that no rapid sedimentation occurs. The experimental settings require the particles to be diluted in the media used to evaluate the cell binding. SEM analysis was also measured on dry samples and gives no information on the colloidal stability.

We agree with the reviewer that aggregation of the microplastic particles would be a problem for our measurements, since the clusters likely would bind differently to cells, and internalization efficiency would be different for clusters compared to single beads. Therefore, we checked for colloidal stability of the beads in cell culture media. We observed no significant aggregation of the beads during cell experiments (lines 151-152, Supplementary Figure 3). Rarely occurring clusters of 2-3 beads were not included in the evaluation, since the particle detection algorithm did not recognize the clusters due to their low cross correlation with a reference bead. Therefore, we are confident that we only analyzed individual beads and no aggregates.

4. it is not clear the choice of the cell type and the link with the eco-tox

We clarified the relevance of murine macrophages as model cells for our study in the introduction (lines 118-123). We focused on J774A.1 murine macrophages as model cells, since in many organ systems, such as the lungs and the gastrointestinal tract, macrophages are among the first cells to encounter inhaled or ingested microplastics. In the gastrointestinal tract, M cells translocate particulate matter, including microplastics, into the Peyer's patches. Resident macrophages then take up the translocated particles and may trigger an immune response. Furthermore, due to the mobility of these cells they can act as transporters for microplastic particles that translocate them into tissues and lead to their distribution in the organism. We analyzed and summarized the role of macrophages for microplastics translocation and distribution in the organism, as well as the mediation of possible effects in a more detailed way in a previous publication (Wieland et al., Journal of Hazardous Materials, 2022).

5. The experimental part is detailed enough. However, the reviewer could not easily find the procedure for the preparation of the working solution of the microparticles for the binding studies, the composition of the media used for the eco-corona (salt water, fresh water) and the sample prep. While the experimental settings are referenced, the main experimental features should be mentioned again. The reviewer also struggles to understand the relevance of the eco-corona when using murine cell lines.

We thank the reviewer for this suggestion. We updated the corresponding parts in the results section (lines 195-197, 279-283, and 309-313).

Exposure to pristine microplastic particles only occurs in very specific settings, e.g., in occupational settings where workers are directly exposed to microplastic dust, such as the flocking industry. However, in most settings, organisms (including mammalian organisms) are exposed to microplastic particles that were in the environment for some time. These particles were subjected to abiotic and biotic weathering, leading to altered surface properties of these particles and the formation of eco-coronas and biofilms. To include such effects in our study, we exposed MM particles to freshwater and salt water and analyzed the eco-corona that formed (lines 153-186, Supplementary Figures 1, 4, and 5, Supplementary Tables 1-3). We showed that environmental exposure altered the particles zeta potential (lines 187-191). This led to changes in their cellular interactions and internalization (lines 224-240, 260-268 and 298-302).

6. the MM zeta potential varied when using different media, the reviewer suggests to further characterising the eco-corona so that it could help understand the zeta-potential change of the eco-corona. the authors refer to a previous publication but, this info should be provided also in this article.

We updated the corresponding paragraph in the manuscript with more detailed information from previous studies we performed to characterize the eco-corona (lines 165-168). Additionally, we identified key microorganisms in the environmental media (Supplementary Figure 4) and performed synchrotron-based scanning transmission X-ray microscopy (STXM) and X-ray photoelectron spectroscopy (XPS) to further quantify components of the eco-corona (lines 168-186, Supplementary Tables 2 and 3, Supplementary Figure 5). We observed an increase in proteins and sugars on the freshwater beads and small amounts of organic nitrogen on the fresh- and saltwater beads that were not detected on pristine MM particles. Organic nitrogen could indicate the presence of biomolecules or other natural organic matter on the particles' surface. Furthermore, we observed an increase in silicon and a decrease in oxygen on the particles' surface, possibly due to interactions with silicic acids. However, small changes in silicon and oxygen signals could not be reliably separated from potential influences of the substrate (thermally oxidized silicon wafer). For the particles incubated in salt water, we additionally observed trace amounts of salts that were also present in the salt water.

7. the internalization experiment needs to be validated with another technique (eg flow cytometry or SPR or QCM) to increase its robustness

We appreciate the suggestion of Reviewer #3 to further validate our internalization experiments. To distinguish between particles that were internalized by a cell and particles that were only attached to the cell surface, we used 3D confocal fluorescence microscopy with a high spatial resolution of 150 nm. We labeled the actin cortex of the cells using phalloidin. In this way, we could decide for each particle individually whether it was inside the cell (surrounded by the actin cortex) or only laying on top. We describe this method more detailed in the Materials & Methods section (lines 711-733). Example images can also be found in our previous publication by Ramsperger et al., Science Advances (2020).

Other methods to quantitatively analyze particle-cell interactions include flow cytometry, surface plasmon resonance spectroscopy (SPR), and quartz crystal microbalance (QCM) assays. However, usually it is very hard to distinguish between adhesion and internalization using these methods. Furthermore, these methods do not provide information about the mechanisms of internalization. Therefore, we decided to validate our internalization assay with living cells transfected with a LifeAct-GFP construct and treated with LysoTracker Red-DND dye (lines 303-323, Supplementary Figure 8).

In this way, we were able to monitor the actin dynamics during internalization, and the lysosomal interactions and acidification of internalized particles. We found that all particles that interacted with lysosomes and became acidified during a measurement showed a substantial increase in actin polymerization in their vicinity during their internalization (Supplementary Figures 9 and 10). These

results show that particles were internalized via the actin-dependent pathways of phagocytosis or macropinocytosis, which validates our previous internalization experiments.

8. The reviewer struggles to appreciate figure 1 as it is too packed and the main results are lost. Is figure 1A a cross section of figure 1B? How was it made? The figure is packed and there are too many “take home messages” which is the variance of the binding affinity when changing the microparticle surface property, surface charge and the presence of eco-corona. The reviewer suggests simplifying it, and moving some data in the SI (eg coverslip binding or similar behaviour of microparticles with same surface charge or same nominal properties).

We thank the reviewer for these suggestions. We updated Figures 1 and 2 accordingly, removed the redundant bar plots, and moved the particle-coverslip binding and adhesion to the Supplementary Figures 6 and 7.

Figure 1A now displays a schematic representation of a trajectory of a single particle during the sedimentation phase. The upper part of the trajectory indicates whether a particle is close to a cell (green parts) or a coverslip (yellow parts). This information was obtained from the classification of the trajectory using the neural network. The lower part of the trajectory shows whether a particle is bound (orange) to a cell/coverslip or diffusing freely (blue parts), depending on its instantaneous velocity $v(t)$. The arrows indicate the instances where a freely diffusing particle bound to a cell/coverslip or where a bound particle detaches from a cell/coverslip. From the number of these binding and unbinding events per second we calculated the binding and unbinding rates k_{on} and k_{off} .

Reviewer #4

There are some nice techniques and methods presented here - I especially like the microfluidic methods to determine the on/off rates and the irreversible binding parameters and the particle uptake analysis.

I do however have a number of major concerns regarding various elements of the work, its overall meaning and the understanding and conclusions being drawn.

(i). Zeta potential is known to change dramatically in serum with hard and soft corona's attached to the particles. The Zeta potential of the particles used here needs to be determined in serum and/or media in which the cells are grown/analysed. I do not see much relevance in equating the Zeta potentials of particles in KCl to how they behave with cells in media (see for example the paper “Surface functionalization affects the zeta potential, coronal stability and membranolytic activity of polymeric nanoparticles in which a series of particles of different Zeta potentials/charges all had the same Zeta potential when in biological solutions).

We thank Reviewer #4 for surveying our manuscript and the helpful comments provided. We agree that incubation in cell culture media can drastically change the particles' zeta potential due to formation of a protein corona and interactions with ions in the media. Therefore, we decided to measure the zeta potential of the microplastic particles before and after an incubation of 2h in cell culture media (lines 145-152, 188-191). As expected, the magnitude of the zeta potential changed, in general becoming less negative (Supplementary Table 1). Interestingly, the differences in the zeta potential were preserved, meaning that particles with an initial zeta potential close to zero were still almost neutral after incubation in cell culture media, whereas particles with a strongly negative zeta potential still had a strongly negative potential after incubation in cell culture media. Overall, the zeta potential of the microplastic particles before and after incubation in cell culture media was strongly correlated (Supplementary Figure 2).

(ii). The corona on particles is typically based on irreversible binding of proteins to give a hard corona and then a more dynamic “soft” corona. How do the particles form a so-called eco-corona? Do the particles “swell” or uptake the buffer/ionic solution? How long would this take ? Would it wash out when the Zeta potentials are measured (this is done in KCl) or when the solvent/buffer is changed? As such I am not convinced by the meaning/measurements of the Zeta potentials of the particles stored in freshwater or salt water and then measured in KCl. I think changes in time of the measurements after removal from freshwater or salt water (i.e. how long they are in the KCl Zeta potential measurement solutions) should be explored.

The structure of an eco-corona is very similar to that of a protein corona. Usually, the first “conditioning layer” of biomolecules forms within seconds after exposure of plastics to the environmental medium (Loeb, Neihof; Adv. Chem; 1975). Over time, a corona forms that can be divided in a hard and soft corona. The hard corona is strongly associated with the microplastic particle, whereas the soft corona is only loosely bound and changes frequently. In a previous study, we could show that this eco-corona has a polymer brush like structure (Witzmann et al., Langmuir, 2022).

Upon change of media (e.g transfer to KCl for zeta potential measurements, to cell culture media for cell experiments, or during the ethanol series for the SEM preparation) the soft corona will likely change. However, we can still observe a visible coating in SEM images, and significant effects of the environmental exposure on the zeta potential and on cellular interactions and internalization. This indicates that a significant part of the eco-corona stays attached even after the media changed.

(iii). If the particles are left in fresh water or salt water for days is there algae growth that generates proteins that bind to the particles? This is quite common in bio-fouling.

Yes, we observed the presence of various organisms in the environmental media (Supplementary Figure 4). Among others, we observed cyanobacteria, diatoms, and green algae of the species *Lagerheimia* in the freshwater samples. In the saltwater samples, we found examples of cyanobacteria. The presence of these microorganisms might explain the protein- and sugar-associated signatures we observed in the synchrotron-based scanning transmission X-ray microscopy (STXM) data (lines 168-176, Supplementary Table 2, Supplementary Figure 5), and the organic nitrogen that we observed with X-ray photoelectron spectroscopy (XPS) on the surface of the environmentally exposed particles (lines 177-186, Supplementary Table 3).

(iv). The beads will be stabilised by surfactants etc.... as the authors say. How stable are these? Are the changes in Zeta potential (e.g. following storage in fresh water) due to the loss of the stabiliser rather than anything else? This is key as the unmodified beads (for example ST) have a Zeta potential that is as -ve as the carboxylate beads which is surprising. In addition, even the unmodified particles might well be stabilised with neutral surfactants such as PVA etc... Is it this that binds to cells or proteins?

We agree with the reviewer that surfactants may play a significant role for the zeta potential of the microplastic particles. With the experiments we performed, we cannot exclude that surfactants were washed out during environmental exposure of the microplastic particles. However, we observed in the scanning electron micrographs that a visible coating forms around the microplastic particles (Supplementary Figure 1, Supplementary Table 1).

To get a better understanding of the eco-corona forming on the microplastic particles, we performed synchrotron-based scanning transmission X-ray microscopy (STXM) and X-ray photoelectron spectroscopy (XPS) to quantify the contents of the eco-corona (lines 168-186, Supplementary Tables 2 and 3, Supplementary Figure 5). We observed small amounts of organic nitrogen, possibly indicating the presence of biomolecules or other natural organic matter, for example humic acids (lines 168-186). Furthermore, we observed an increase in the amount of silicon and a decrease in the amount of oxygen

on the particles' surface. However, small changes in silicon and oxygen signals could not be reliably separated from potential influences of the substrate (thermally oxidized silicon wafer). Additionally, we observed trace amounts of salts on the particles exposed to salt water.

We assume that this eco-corona carries negative charges, as most natural organic matter, such as humic acids, is negatively charged (lines 382-385). This is consistent with our observation that the zeta potential of environmentally exposed microplastic particles was more negative than the zeta potential of the respective pristine particles.

(v). As a general rule +vely charged particles/molecules interact with cells much more readily than -vely charged ones (due to the nature of the cell membrane) - This is opposite to that seen here. Please explain.

We thank the reviewer for this remark. The exact role of the charge polarity for microparticle-cell interactions is not unanimously clear. Since cells are in general on average negatively charged, it is expected that microparticles with a positive surface charge interact more readily with the cells, as pointed out by the reviewer. However, there are studies that show that both negatively and positively charged microparticles are internalized by cells efficiently, and the internalization efficiency increases with an increase in negative surface charge. We elaborated this in the introduction (lines 88-96).

However, we agree with the reviewer that we only had one particle type with an overall positive zeta potential, making it difficult to draw robust conclusions about the role of charge polarity for cellular interactions and internalization. In the previous version of the manuscript, we included 3 different functionalized particles, next to the positive amine-modified particles there were two carboxylated particle types with a negative zeta potential. These functionalized particles behaved very similar to the unmodified particles. Functionalized particles that were strongly charged interacted strongly with the cells and were more frequently internalized, following the correlation between particle-cell binding, adhesion, absolute internalization probability and the zeta potential. However, we decided to leave them out of this study for several reasons.

First, as pointed out by the reviewer, only one of these functionalized particles had a positive zeta potential. Therefore, it is very difficult to draw statistically robust conclusions about the role of charge polarity for cellular interactions and internalization. Secondly, the inclusion of more functionalized particle types would substantially increase the complexity of our study, since particle-cell interactions are likely influenced by other factors beyond just charge, for example the chemical identity of the functional groups that could lead to more specific receptor-ligand interactions.

We believe that it will be interesting to investigate the effect of different charge polarities and functional surface groups of microplastic particles for their interactions with environmental and biological media, and their cellular interactions and internalization. This might also be relevant in an ecotoxicological context where environmental plastic particles carry functional surface groups, e.g., due to environmental weathering. However, we think that this falls outside of the scope of this study, which primarily focuses on the role of the zeta potential for particle-cell interactions of nominally identical, unmodified model microplastic particles.

The whole section on their claims that local electrostatic interactions and binding with the cell membrane is flawed - the Zeta potential (in media) has not been determined but I would imagine they will be pretty similar across all the particles.

To better understand the mechanisms behind the particle-cell interactions, we measured the zeta potential of the microplastic particles before and after incubation in cell culture media (lines 145-152 and 188-191, Supplementary Table 1). These measurements show that the magnitude of the zeta

potential became smaller (Supplementary Table 1). However, the differences in the zeta potential were preserved, meaning that particles with an initial zeta potential close to zero were still almost neutral after incubation in cell culture media, whereas particles with a strongly negative zeta potential still had a strongly negatively potential after incubation in cell culture media (Supplementary Figure 2). The particles incubated in cell culture media still spanned a wide range of zeta potentials from -6.2 mV to -41.5 mV. Therefore, we think that electrostatic interactions together with the possible charge heterogeneity on cell surfaces and the presence of multivalent ions in the media contributed to particle-cell binding and adhesion (lines 399-408).

(vi). The fact that cell binding and coverslip binding show the same trends and similar on/off rates for the particles worries me. What is the nature of interaction on the coverslips? Is the particle just getting “stuck” in defects on the cover slip surface? i.e. they say “particles adhered strongly to cells and coverslips...”. What is the nature of the coverslip, is it getting coated with protein too? Is this what is controlling particle binding?

Similar to cells, glass coverslips carry negative surface charges. During sample preparation and experiments, the coverslips were exposed to cell culture media for several hours, which potentially led to the adsorption of proteins and amino acids from the media that carry negative as well as positive charges. Therefore, we assume that the coverslips likely were charged heterogeneously. Together with the presence of multivalent ions in the media, electrostatic interactions between particles and coverslips might therefore mediate particle-coverslip binding and adhesion.

Furthermore, we observed that particle-coverslip adhesion generally was weaker than particle-cell adhesion (lines 216-223, 237-240, 241-247, 257-259, 266-268, and 269-275, Supplementary Figures 6 and 7). We think that this might be caused by the different surface geometries of cells and coverslips. Since cells are softer and more irregularly shaped, the contact area to the microplastic particles might be larger, compared to the smooth and flat surface of the coverslips.

(vii). Figure 3. The sign of the Zeta potential on the particles (e.g. +29 mV for the amine (+vely charged particles and -91 mV for the negatively acid particles) is ignored in the graphs. I do not think it is OK to put +ve and -ve Zeta values on a log plot and hence ignore the sign. Cells respond so differently to different charged particles. Of course, they could all have the same Zeta potential (due to a protein corona) but then we would expect to see few differences between the particles.

We thank the reviewer for this suggestion. Since we decided to leave out the intentionally functionalized particles in this study as explained in the paragraph (v), also the positively charged amino-modified particles are no longer a part of this study. The remaining unmodified particles were all negatively charged. Therefore, we no longer ignore the sign of the charge in the graphs.

We agree with the reviewer that cells may respond differently to charges of different polarity and different chemical surface groups. Therefore, we think that it may be interesting to investigate the effect of different functional surface groups and charge polarities of microplastic particles on their interactions with environmental and biological media, and their cellular interactions and internalization. However, we believe that this is outside of the scope of this study.

(viii). Figure 3 is a log/log plot. As such I am not convinced that there is any relationship here and I am not sure why a log scale is being used or relevant.

To show that there is indeed a correlation between the binding kinetics, adhesion forces, and absolute internalization probability (Figure 3) and the zeta potential, we performed Pearson's R test. In all cases, the correlations were highly significant ($p < 0.001$) and strong ($|R| > 0.8$).

Currently, we present Figure 3 B, as well as Figure 1 E, and Figure 2 B as a logarithmic plot. The log-plots help to display the data in a way that resolves the small differences between the particles with a zeta potential close to zero, while at the same time making it possible to show the strong interactions and internalization of strongly negatively charged particles. Other plots, like Figure 1 C and D, were displayed as a log-log plot. This additionally may help to understand the underlying relationship between the binding kinetics and the zeta potential. In general, power laws (e.g., a polynomial function) appear as a straight line in a log-log plot. Therefore, the data in Figure 1 C and D following approximately a straight line could indicate that the binding kinetics and the zeta potential of the microplastic particles are linked by such a power law.

Minor general points

Label the particles 1-11.... Having a list of abbreviations helps little and makes it very cumbersome.

We appreciate this suggestion. However, we think that it is more intuitive for many people to associate the particle types with the names of the manufacturers. Additionally, the list of abbreviations helps to clarify the history of the environmentally exposed beads. For example, the abbreviation “MM-SW2” makes it immediately clear that these are MM particles that were incubated in salt water (SW) for 2 weeks. Therefore, we decided to keep the abbreviations.

The density of the particles needs to be measured as this might well also affect binding.

We thank the reviewer for pointing out this possibility. Data about the density of the microplastic particles was available from most manufacturers, except Polysciences, Kisker, and Tianjin. The remaining particle types had densities in a very narrow range from 1.03 g/cm³ (micromod) to 1.06 g/cm³ (Phosphorex). We provided the values in Supplementary Table 1.

However, we are not convinced that these small differences in densities affect the binding or adhesion of the microparticles. A spherical particle with a diameter of 3 μm and a density of 1.05 g/cm³ in water (density 1 g/cm³) would experience a gravitational force of approximately 6.9 x 10⁻³ pN pressing the particle down on the cell surface. This force is four orders of magnitude lower than the force we exert during the flushing phase. Thus, gravitational pull likely does not substantially contribute to the adhesion of microplastic particles to cells, but rather electrostatic forces and receptor-ligand interactions. Thus, even larger differences in the particles’ density would likely not affect the particle-cell binding and adhesion.

Use of the word “plain” for particles - “unmodified” would be better.

We thank the reviewer for this suggestion. We replaced “plain” with “unmodified”, since only 2 of the 8 manufacturers of microparticles used in this study designated their unmodified particles as “plain” (Polysciences, micromod).

Please don't repeat and over-labor trivial points e.g. kon/koff etc...

We shortened the respective parts in the manuscript and formulated the relevant information about k_{on} and k_{off} in a more concise way (e.g., lines 245-247, caption of Figure 1).

Particle clumping behavior should be mentioned - often an issue with low Zeta potential particles.

We agree with the reviewer that clustering of the microplastic particles would be a problem for our measurements. However, we observed no significant clustering of the beads in the cell culture media during cell experiments (lines 151-152, Supplementary Figure 3). Aggregates of 2-3 beads occurred only very rarely. These were not included in the evaluation, since the particle detection algorithm did

not recognize the clusters due to their low cross correlation with a reference bead. Therefore, we are confident that we only analyzed individual beads and no aggregates.

SI: Images for MM-SW2/4 show salt crystals. I would suggest this is due to lack of washing (as the beads have been soaked in salt water) rather than a real feature of the beads.

We appreciate this observation of the reviewer. SEM images in Supplementary Figure 1 were acquired at a voltage of 5 kV using an Everhart-Thornley detector. Consequently, mostly secondary electrons were observed, and backscattered electrons only minimally contributed to the signal, leading to no or only minimal material contrast in the SEM images. Therefore, it is difficult to identify the components of the eco-corona observed in the SEM images. Furthermore, preparation of the samples for SEM imaging required multiple changes of the media for the fixation step and the subsequent dehydration of the sample via an ethanol series. It is expectable that during these media changes ions were removed from the samples, so that salt crystals could not built up during drying of the samples.

To identify the constituents of the eco-corona observed in the SEM images, we additionally performed STXM and XPS measurements. In the XPS measurements (Supplementary Table 2) we observed organic nitrogen on the saltwater beads, indicating the presence of biomolecules or other natural organic matter, for example humic acids (lines 177-186). Furthermore, we observed an increase in the amount of silicon and a decrease in the amount of oxygen on the particles' surface. Furthermore, there were there were trace amounts of salts on the particles exposed to salt water.

REVIEWERS' COMMENTS

Reviewer #2 (Remarks to the Author):

I have re-read this article carefully. think the authors have done their best to reflect on the issues, and to be open minded. Not all of the points of all the reviewers could be answered, but many were.

The issue of allowing space to breathe for new communities is important in judging this issues.

I would accept this manuscript. I think it will be interesting to and carry valuable messages onwards to new community.

Reviewer #3 (Remarks to the Author):

The reviewer feels that the article is significantly improved and it is now suitable for publication

The choice of model microplastics strongly affects particle-cell interactions – Letter of Response

Reviewer #2:

I have re-read this article carefully. think the authors have done their best to reflect on the issues, and to be open minded. Not all of the points of all the reviewers could be answered, but many were.

The issue of allowing space to breathe for new communities is important in judging this issues.

I would accept this manuscript. I think it will be interesting to and carry valuable messages onwards to new community.

We thank Reviewer #2 for the efforts to review our manuscript. The constructive comments provided during the review process significantly contributed to improving our research article.

Reviewer #3:

The reviewer feels that the article is significantly improved and it is now suitable for publication.

We appreciate the thorough examination of our manuscript and thank Reviewer #3 for the efforts. We substantially improved our manuscript based on the helpful comments of the reviewer during the review process.